

# Instabilities of quantum critical metals in the limit $N_f \to 0$

Petter Säterskog[1,2⋆]

**1** Blekingegatan 27, 118 56 Stockholm, Sweden
**2** Nordita, KTH Royal Institute of Technology and Stockholm University,
Roslagstullsbacken 23, SE-106 91 Stockholm, Sweden

⋆ petter.saterskog@gmail.com

## Abstract

We study a model in 2+1 dimensions composed of a Fermi surface of $N_f$ flavors of fermions coupled to scalar fluctuations near quantum critical points (QCPs). The $N_f \to 0$ limit allows us to non-perturbatively calculate the long-range behavior of fermion correlation functions. We use this to calculate charge, spin and pair susceptibilities near different QCPs at zero and finite temperatures, with zero and finite order parameter gaps. While fluctuations smear out the fermionic quasiparticles, we find QCPs where the overall effect of fluctuations leads to enhanced pairing. We also find QCPs where the fluctuations induce spin and charge density wave instabilities for a finite interval of order parameter fluctuation gaps at $T = 0$. We restore a subset of the diagrams suppressed in the $N_f \to 0$ limit, all diagrams with internal fermion loops with at most 2 vertices, and find that this does not change the long-range behavior of correlators except right at the QCPs.



# 1  Introduction

Interacting fermions at finite density form a rich physical system that can be used to describe many condensed matter phenomena. Such systems are generally governed by the Fermi liquid IR fixed point but are susceptible to several symmetry breaking instabilities resulting in states with superconductivity (SC), ferromagnetism (FM), a Fermi surface breaking rotational symmetry, or a state with charge or spin density waves (CDW, SDW) breaking translational symmetry.

Transitions to these states can sometimes be tuned to zero temperature in condensed matter systems by changing e.g. pressure or the concentration of different dopants. When such a transition is second order, it is associated with strong fluctuations of the order parameter field near the zero temperature quantum critical point (QCP). The interaction between these order parameter fluctuations and the fermions is relevant in two dimensions and may thus alter the Fermi liquid IR fixed point. More dramatically, the interaction with near-critical fluctuations may themselves induce an instability to a symmetry broken state. This makes interacting fermions in two dimensions near a QCP an even richer system to study, we refer to them as quantum critical metals. The modified fixed points are called non-Fermi liquids (NFLs) or strange metals. The sharp quasiparticles of the Fermi liquid are destroyed but the Fermi surface remains, they are still metallic but their scaling laws are modified.

Many experimental systems of interacting fermions that do not fit into a Fermi liquid description are in fact (quasi) two-dimensional and close to QCPs. Examples are cuprate [1, 2] and iron pnictide [3–5] superconductors showing NFL physics in the normal phase and critical temperatures higher than what can be explained within Fermi-liquid theory. Critical fluctuations can lead to NFL physics but the destruction of the quasiparticles is thought to limit pairing. At the same time, certain critical fluctuations can work as a pairing glue that enhances superconductivity. This makes it important to study quantum critical metals to understand which of these competing effects dominates and whether the vicinity of the QCPs can explain

the enhanced superconductivity.

A fermion-boson model [6–8] is typically employed to study quantum critical metals. The fermions represent low-energy excitations at a Fermi surface and the boson takes the role of the critical fluctuations of the order parameter. The symmetry broken at the QCP is reflected in the symmetry of the boson. There is an important distinction between transitions breaking translational symmetry where the order parameter fluctuations have a finite momentum $Q$ and those that do not break translational symmetry and the order parameter fluctuations are centered around $Q = 0$. We only consider the $Q = 0$ case in this work. As mentioned, these are strongly coupled systems in two dimensions and the fermion-boson model is thus not amenable to perturbation theory. As a consequence, several approximation methods have been employed. A common approach has been to extend the theory to get a new expansion parameter that is taken to a limit where we can do calculations. The resulting system is not the one we are ultimately interested in, but the hope is that some of the interesting physics remains. One example of this is to not study the model in 2 dimensions, but in $d_c - \epsilon$ dimensions [9–13]. $\epsilon$ is treated as a small perturbation away from the upper critical dimension $d_c$ where the theory can be studied perturbatively.

Another approach is to change the field content of the model. This can be done in several different ways. Instead of considering just a single fermionic field we may consider $N_f$ flavors of fermions all coupling to the same bosonic field with a rescaled coupling constant $g = \lambda / \sqrt{N_f}$. This extended model has been studied in the limit of large $N_f$ [14, 15]. It was later found that the scaling laws presented in these works do not survive when adding higher loop corrections [16, 17].

A related approach is the matrix large-$N$ limit in which the boson is an $N \times N$ component matrix transforming in the adjoint representation of a global $SU(N)$ symmetry group and the fermion an $N$ component vector transforming in the fundamental. The coupling is scaled as $g = \lambda / \sqrt{N}$ and the theory is studied in the limit of large $N$. All fermion loops are suppressed in this limit as well as all non-planar diagrams. This means that the boson receives no corrections from the fermion and only a subset of the boson corrections to the fermion are kept. Polynomially many diagrams contribute at each order as opposed to the factorially many in the full theory. Both non-Fermi liquid physics [18–21] and the superconducting instability [22–25] of quantum critical metals have been studied in this limit.

Finally we arrive at the small $N_f$ limit which is the approach considered in this paper. This is set up similar to the large $N_f$ limit, but we do not rescale the coupling constant and we bring $N_f$ to 0 instead of infinity. It may seem awkward to set $N_f$ to zero, in the large $N_f$ and $N$ cases above we have a sequence of physical theories with integer numbers of fields that approach the limit under study. Here there is no such sequence of physical theories but in practice there is no big difference. All diagrams come with an order of $N_f$ and we keep the lowest $N_f$ order instead of the highest in $N$. It turns out that the $N_f \to 0$ limit keeps all of the diagrams kept in the matrix large-$N$ limit and it additionally keeps the crossed diagrams and thus contains factorially many diagrams at each order. Only the fermionic loops are suppressed by taking $N_f \to 0$. This large set of included diagrams is unique to the small $N_f$ limit among the analytical approaches. This may allow the small $N_f$ limit to uncover non-perturbative phenomena that are otherwise not found.

Intuitively, the small $N_f$ limit can be understood as the fermions living in a background of a fluctuating bosonic field $\phi$ that gives the fermions corrections at all orders of the coupling constant, but all the corrections from the fermions onto the field $\phi$ are turned off and $\phi$ behaves as if it were free.

The momentum space retarded two-point function was calculated in the small $N_f$ limit in [26]. Surprisingly, it was possible to obtain an (almost) closed form expression, and it showed that the fermion dispersion becomes non-monotonic due to corrections from the criti-

cal fluctuations. This was then extended in [27] where a framework was developed to calculate general fermion $n$-point correlation functions in the small $N_f$ limit of a fermion-boson model.

In addition to keeping a superset of all diagrams of the matrix large $N$ limit, the small $N_f$ limit allows us to calculate explicit correlation functions, albeit in real space and for distances longer than $1/k_F$. This provides a useful complement to the results that have been obtained using other limits that are mostly in the form of scaling laws and renormalization group flows.

The small $N_f$ calculations are done in a long distance limit where we consider correlators between local operators separated far compared to the inverse Fermi momentum. This low-energy limit does not commute with either the matrix large-$N$ limit nor the small $N_f$ limit. An important effect that is missed by taking these limits before the low-energy limit is called Landau-damping. The boson gets dressed by fermion bubbles that change the IR scaling of the boson. There are earlier works in the small $N_f$ limit where the authors incorporated these effects from the start by using an explicitly Landau-damped boson when calculating the real space fermion two-point function [8, 28–31]. This takes into account some of the low-energy effects missed in taking the $N_f \rightarrow 0$ limit first. However, this was not done in a systematic way. Landau-damping effects were included systematically (with further constraints on the order of other limits involved in the definition of the theory) by considering the particular limit $N_f \rightarrow 0$, $k_F \rightarrow \infty$, $N_f k_F = $ constant in [32]. This limit does not go as far as taking $k_F \rightarrow \infty$ first, but it gives a regime where we get Landau damping effects, while still allowing us to use the methods of the strict small $N_f$ limit to calculate correlation functions.

The intuition is now slightly changed, the bath the fermions live in receives some corrections from the fermions such that its IR scaling is changed. However, the corrections are only to the boson two-point function and the bosonic fluctuations remain Gaussian. The momentum space fermion two-point function was calculated in this limit in [32] and it was found that the non-monotonicity of the fermion dispersion disappeared for a non-zero $N_f k_F$.

The above works in the small $N_f$ limit studied non-Fermi liquid physics of quantum critical metals in the normal phase. The cuprates and iron pnictides additionally show enhanced superconductivity compared to what would be expected from Fermi liquid theory and this has not yet been studied in the small $N_f$ limit. In this paper we explore how critical fluctuations can lead to instabilities of finite density fermions in the small $N_f$ limit. We do this by using the framework developed in [27] to calculate spin, charge and pair correlation functions. As opposed to the fermion self-energy studied in earlier works, these higher order $n$-point functions are sensitive to the symmetry of the fermion-boson coupling so we consider different types of QCPs and additionally fermions coupled to a $U(1)$ gauge field which can be described by the same fermion-boson model. Experimental systems are always at a finite temperature, and it has been pointed out that non-Fermi liquids may behave quite differently at finite temperatures compared to at $T = 0$ [25, 33–35]. Because of this, we go beyond earlier small $N_f$ works by working at both $T = 0$ and finite $T$. Additionally, we depart from the critical point and also consider gapped order parameter fluctuations to obtain a full phase diagram on the disordered side of the QCPs.

We give a brief summary of the main results here. We show that the small-$N_f$, long-distance, real space correlation functions can be written as the free correlators, multiplied by the exponential of a linear combination of two functions we call $h^+(\tau, x)$ and $h^-(\tau, x)$. The former of these functions captures processes with fermions interacting with fermions within the same Fermi surface patch and is responsible for quasiparticle smearing. The latter captures processes where fermions interact with fermions in an antipodal patch and is responsible for attraction and repulsion between antipodal fermions. The coefficients of the linear combination of $h^\pm(\tau, x)$ in the exponents are given by the coupling function in the different Fermi surface patches. $h^+(\tau, x)$ shows up in fermion self-energy corrections whereas both $h^+(\tau, x)$ and $h^-(\tau, x)$ show up in charge, spin and pair correlators. The $h^\pm(\tau, x)$ functions

are both obtained from an integral over the boson propagator and we find a general inequality, $|h^-(\tau, x)| \geq |h^+(\tau, x)|$, meaning that the effects of fermions interacting between antipodal patches are stronger than the intra-patch interactions, except when the inequality is saturated. $h^-(\tau, x)$ grows unboundedly in $x$ and depending on the signs of the couplings, this can result in diverging susceptibilities. The above inequality is saturated at finite temperatures, resulting in finite susceptibilities, but we find that the charge/spin susceptibilities diverge at zero temperature up to a critical order parameter gap, for certain couplings. We interpret this as an instability towards charge/spin density waves at zero temperature, with a new induced QCP at the critical gap. We also find an enhanced/cured divergence of the pair-susceptibility as temperature is lowered, again depending on the signs of couplings. We change the boson propagator to include Landau damping as was done in [32], and find that the long-distance behavior of correlation functions is unchanged, except right at the QCP.

The paper is organized as follows. In Section 2 we present the model we have studied and the particular limit this is studied in. We start out with a very general model to first explore the possible phases without restrictions. In this section we also extend the framework in [27] to account for momentum dependent couplings and finite temperature. In Section 3 we calculate pair, charge and spin correlation functions of the general model in search of unstable modes. We consider the effect of Landau-damping corrections by redoing this in the double limit of [32] in Section 4. In Section 5 we apply these results to specific physical systems that can be described by the model that we have been exploring. We consider charge and spin nematic transitions, ferromagnetic transition, circulating current transition and coupling to an emergent $U(1)$ gauge field, in different subsections. We also consider the case of fermions coupled to multiple near-critical fluctuations. Each of these subsections are ended with a discussion where we compare our results to earlier findings on similar models. Finally in Section 6 we summarize our work and pose some open questions for the future.

## 2 Setup and calculation of fermion $n$-point functions in the $N_f \to 0$ limit

Similarly to [15, 23] we consider a general theory that has applications to several systems composed of a Fermi surface coupled to $Q = 0$ gapless bosonic excitations. The theory contains free parameters whose values depend on the particular application we consider. For now, we keep the values of these parameters unspecified. In Section 5 we consider physical realizations of this model with more concrete parameters.

We consider a Fermi surface of $N_f$ flavors of spin-1/2 fermions $\psi_{i,\sigma}$ and order parameter fluctuation fields $\phi_a$:

$$S = \int d\tau d^2x \left[ \psi_{i,\sigma}^\dagger (\partial_\tau + \epsilon(i\nabla) - \mu) \psi_{i,\sigma} + \frac{1}{2}(\partial_\tau \phi_a)^2 + \frac{c_a^2}{2}(\nabla \phi_a)^2 + \frac{r_a}{2}\phi_a^2 - \phi_a(x)O_a(x) \right]. \tag{1}$$

$i$ is the fermion flavor index and takes values $i = 1, ..., N_f$, $\sigma$ is the fermion spin $\sigma = \uparrow, \downarrow$. The fermions transform in the fundamental representation of a global $U(N_f)$ flavor symmetry group. $a = 1, ..., N_b$ enumerates the order parameters/order parameter components. $r_a$ is a tuning parameter across the QCP where $\langle \phi_a \rangle \neq 0$ for $r_a < 0$ and $\langle \phi_a \rangle = 0$ for $r_a > 0$. For simplicity, we limit ourselves to the case of a circular Fermi surface $\epsilon(\mathbf{k}) = k^2/2m$. We define the bare Fermi momentum $k_F = \sqrt{2m\mu}$ and bare Fermi velocity $v_F = \sqrt{2\mu/m}$. The fields $\phi_a$

couple to the operator:

$$O_a(x) = \sum_{i=1,\sigma=\uparrow,\downarrow}^{N_f} \int \frac{\mathrm{d}^3k\mathrm{d}^3q}{(2\pi)^6} \lambda_{a,\sigma}(\mathbf{k})\psi^\dagger_{\sigma,i}\left(k-\frac{q}{2}\right)\psi_{i,\sigma}\left(k+\frac{q}{2}\right)e^{iqx}, \tag{2}$$

where we use the notation $kx = -\omega\tau + \mathbf{k}\cdot\mathbf{x}$ and the Fourier transformed fields:

$$\psi(x) = \int \frac{\mathrm{d}^3k}{(2\pi)^3}e^{ikx}\psi(k) \tag{3}$$

$$\psi^\dagger(x) = \int \frac{\mathrm{d}^3k}{(2\pi)^3}e^{-ikx}\psi^\dagger(k). \tag{4}$$

The coupling function $\lambda_{a,\sigma}(\mathbf{k})$ characterizes how the fermion couples to the order parameter fluctuations. Only the dependence on the *direction* of $\mathbf{k}$ is relevant for low energy excitations close to the Fermi surface. A $\phi^4$ term can be consistently left out since it is not generated in the $N_f = 0$ theory[1]. We consider dynamic bosons but the static case of our results is obtained by the definitions

$$\phi_a = \tilde{\phi}_a/c_a$$
$$\lambda_{a,\sigma}(\mathbf{k}) = c_a\tilde{\lambda}_{a,\sigma}(\mathbf{k})$$
$$r_a = c_a^2\tilde{r}_a, \tag{5}$$

and the limit $c_a \to \infty$.

With an eye towards finding instabilities to pairing and charge/spin order we consider correlation functions of pair creation/annihilation operators and charge/spin density operators:

$$b(x) = \sum_{i=1}^{N_f} \psi_{i,\uparrow}(x)\psi_{i,\downarrow}(x) \tag{6}$$

$$b^\dagger(x) = \sum_{i=1}^{N_f} \psi^\dagger_{i,\downarrow}(x)\psi^\dagger_{i,\uparrow}(x) \tag{7}$$

$$\rho_c(x) = \rho_\uparrow(x) + \rho_\downarrow(x) \tag{8}$$

$$\rho_s(x) = \rho_\uparrow(x) - \rho_\downarrow(x), \tag{9}$$

where

$$\rho_\sigma(x) = \sum_{i=1}^{N_f} \psi^\dagger_{i,\sigma}(x)\psi_{i,\sigma}(x). \tag{10}$$

$\rho_\sigma$ is invariant under the global $U(N_f)$ whereas $b$ is not. We consider real space correlators of these operators:

$$\langle b^\dagger(0)b(x)\rangle, \ \langle\rho_c(0)\rho_c(x)\rangle, \ \langle\rho_s(0)\rho_s(x)\rangle.$$

Writing the charge and spin correlators in terms of $\rho_\uparrow, \rho_\downarrow$ correlation functions we have:

$$\langle\rho_{c,s}(0)\rho_{c,s}(x)\rangle = \sum_\alpha \langle\rho_\alpha(0)\rho_\alpha(x)\rangle \pm 2\langle\rho_\uparrow(0)\rho_\downarrow(x)\rangle. \tag{11}$$

---

[1]The $\phi^4$ term is relevant in the strict $N_f = 0$ theory but the interaction is irrelevant (at $T = 0$) in the more physically interesting case of a Landau-damped boson.

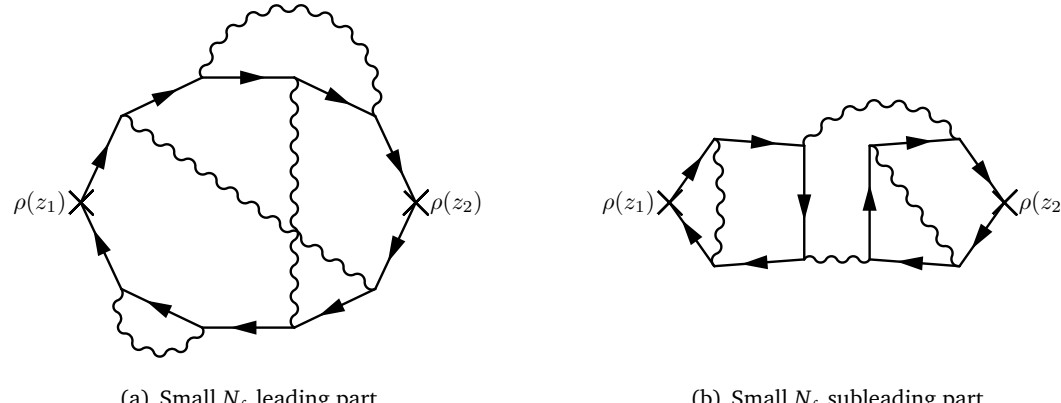

(a) Small $N_f$ leading part        (b) Small $N_f$ subleading part

Figure 1: Once the fermionic fields have been integrated out and the resulting determinant set to 1 (by the small $N_f$ limit) there are two classes of diagrams contributing to the fermion density-density correlator. (a) Shows one of the diagrams in the first class that contributes at order $N_f$ (b) Shows a diagram in the second class that contributes at order $N_f^2$.

The above operators all contain two fermionic fields. This, together with the $N_f \to 0$ limit, means that order by order the contributing diagrams only contain two continuous fermionic lines, each starting and ending at the above bilinears. There is only one way of connecting these lines in the case of the pair operators and the $\langle \rho_\uparrow(0)\rho_\downarrow(x) \rangle$ correlator but for the $\langle \rho_\alpha \rho_\alpha \rangle$ correlator these lines can be connected in two different ways. The fermionic lines attached to the density operators can either connect back to the same operator or connect the other insertion, see Fig. 1. We get an additional sum over fermionic flavors when the propagators connect back to the same insertion as in Fig. 1(b). This means that those contractions are subleading in the small $N_f$ limit. The $\langle \rho_\uparrow \rho_\downarrow \rangle$ correlator only has that type of contraction and will always be subleading:

$$\langle b^\dagger(0)b(x) \rangle = \mathcal{O}(N_f) \tag{12}$$

$$\langle \rho_\sigma(0)\rho_\sigma(x) \rangle = \mathcal{O}(N_f) \tag{13}$$

$$\langle \rho_\uparrow(0)\rho_\downarrow(x) \rangle = \mathcal{O}(N_f^2). \tag{14}$$

We can thus not differentiate between charge and spin density fluctuations at leading order in the $N_f \to 0$ limit. Let us now see how we can calculate these correlation functions. We start off by writing a generating functional with sources $J_{\sigma,i}, J_{\sigma,i}^\dagger$ for the fermionic fields:

$$Z[J^\dagger, J] = \int \mathcal{D}\psi^\dagger \mathcal{D}\psi \mathcal{D}\phi \exp\left(-S[\psi^\dagger, \psi, \phi] - \int d^3z(J_{\sigma,i}\psi_{\sigma,i}^\dagger + J_{\sigma,i}^\dagger \psi_{\sigma,i})\right). \tag{15}$$

Next we integrate out the fermionic fields. To do that we write out the interaction in terms of real space fermionic fields:

$$\int d^3x \, \phi_a(x) O_a(x) = \sum_{i=1,\sigma=\uparrow,\downarrow}^{N_f} \int \frac{d^3k \, d^3x_1 d^3x_2}{(2\pi)^3} e^{ik(x_1-x_2)} \lambda_{a,\sigma}(\mathbf{k}) \psi_{\sigma,i}^\dagger(x_1) \psi_{\sigma,i}(x_2) \phi_a\left(\frac{x_1+x_2}{2}\right). \tag{16}$$

The fermionic integral is Gaussian and the generating functional is obtained as

$$Z[J^\dagger, J] = \int \mathcal{D}\phi \exp\left(-S_{\det}[\phi] - S_b[\phi] - \int d^3z \, d^3z' J_{\sigma,i}^\dagger(z) G_{\sigma\sigma',ij}[\phi](z,z') J_{\sigma',j}(z')\right). \tag{17}$$

The background field Green's function is diagonal in spin and flavor indices and independent of flavor so we can write it as $G_{\sigma\sigma',ij}[\phi](z,z') = \delta_{\sigma\sigma'}\delta_{ij}G_\sigma[\phi](z,z')$. The background field Green's function is the solution to

$$\left(-\partial_{\tau_1} - \epsilon(-i\nabla_1) + \mu\right)G_\sigma[\phi](x_1,x_2) + \tag{18}$$

$$\sum_{a=1}^{N_b}\int\frac{d^3k\, d^3x'}{(2\pi)^3}e^{ik(x_1-x')}\lambda_{a,\sigma}(\mathbf{k})\phi_a\left(\frac{x_1+x'}{2}\right)G_\sigma[\phi](x',x_2) \tag{19}$$

$$= \delta^3(x_1 - x_2), \tag{20}$$

and

$$S_{\text{det}}[\phi] = -\text{tr}\log G_{\sigma\sigma',ij}[\phi](z,z') = -N_f\sum_\sigma \text{tr}\log G_\sigma[\phi](z,z'). \tag{21}$$

In momentum space:

$$G(x_1,x_2) = \int\frac{d^3k_1 d^3k_2}{(2\pi)^6}e^{ik_1x_1 - ik_2x_2}G(k_1,k_2), \tag{22}$$

we have

$$\left(i\omega_1 - \epsilon(k) + \mu\right)G_\sigma[\phi](k_1,k_2) + \sum_{a=1}^{N_b}\int\frac{d^3q}{(2\pi)^3}\lambda_{a,\sigma}\left(\mathbf{k}_1 - \frac{\mathbf{q}}{2}\right)\phi_a(q)\,G_\sigma[\phi](k_1-q,k_2)$$
$$= (2\pi)^3\delta^3(k_1-k_2). \tag{23}$$

Following the procedure of [27] we expand around a point $\hat{n}k_F$ on the Fermi surface. Now we additionally expand the functions $\lambda_{a,\sigma}(\mathbf{k}) = \lambda_{a,\sigma,\hat{n}} + \mathcal{O}(\mathbf{k} - \hat{n}k_F)$, and Fourier transform back to real space:

$$\left(-\partial_{\tau_1} + v_F(i\hat{n}\cdot\nabla_1 + k_F) + \sum_{a=1}^{N_b}\lambda_{a,\sigma,\hat{n}}\phi_{a,\sigma}(x_1)\right)G_{\sigma,\hat{n}}[\phi](x_1,x_2) = \delta^3(x_1-x_2). \tag{24}$$

The Euclidean time variables $\tau_1$ and $\tau_2$ are constrained to the interval $(0,\beta)$ when the theory is considered at a finite temperature. The fermionic Green's function $G_\sigma[\phi]$ has antiperiodic boundary conditions and the bosonic fields $\phi_a$ have periodic boundary conditions in the $\tau$ direction. The solution to Eq. (24) with antiperiodic boundary conditions is given by

$$G_{\sigma,\hat{n}}[\phi](x_1,x_2) = f_{\hat{n}}^-(x_1-x_2)\exp\left(ik_F\hat{n}\cdot(\mathbf{x}_1-\mathbf{x}_2) + I_{\sigma,\hat{n}}[\phi](x_1,x_2)\right), \tag{25}$$

where

$$I_{\sigma,\hat{n}}[\phi_a](x_1,x_2) = \sum_{a=1}^{N_b}\lambda_{a,\sigma,\hat{n}}\int d^3x\,\phi_a(x)\left(f_{\hat{n}}^+(x-x_1) - f_{\hat{n}}^+(x-x_2)\right), \tag{26}$$

and $f_{\hat{n}}^\pm(x)$ are the $\beta$-periodic[2] (+) and $\beta$-antiperiodic (−) (in $\tau$) solutions of

$$\left(-\partial_{\tau_1} + iv_F\hat{n}\cdot\nabla\right)f_{\hat{n}}^\pm(x_1,x_2) = \delta^3(x_1-x_2). \tag{27}$$

---

[2]Note that in finding the solution for the fermionic background field Green's function we make use of Eq. (28) which is a Green's function for the free fermion (in patch-coordinates) but with boson statistics. The method we employ here is referred to as functional bosonization in some of the earlier works [28].

These functions can be expressed as:

$$f_{\hat{n}}^{+}(x) = \frac{T \cot\left(\pi T \left(\frac{i\hat{n}\cdot\mathbf{x}}{v_F} - \tau\right)\right)}{2v_F} \delta(\hat{n} \times \mathbf{x}) \tag{28}$$

$$f_{\hat{n}}^{-}(x) = \frac{T \csc\left(\pi T \left(\frac{i\hat{n}\cdot\mathbf{x}}{v_F} - \tau\right)\right)}{2v_F} \delta(\hat{n} \times \mathbf{x}). \tag{29}$$

We define the cross product as $\mathbf{a} \times \mathbf{b} = a_x b_y - a_y b_x$. The found solution $G_{\sigma,\hat{n}}[\phi](x_1, x_2)$ is only valid when acting on momentum modes close to $k_F \hat{n}$. As in [27] we project onto these modes and integrate over different directions $\hat{n}(\theta)$ to obtain an operator $G_{\text{IR}}[\phi](x_1, x_2)$ that is valid as long as $|\mathbf{x_2} - \mathbf{x_1}| \gg k_F^{-1}$:

$$\begin{aligned}
G_{\text{IR},\sigma}[\phi](x_1, x_2) &= \int d^2\mathbf{x}' \int \frac{dk k d\theta}{(2\pi)^2} f_{\hat{n}(\theta)}^{-}(\tau_1 - \tau_2, \mathbf{x}_1 - \mathbf{x}') \times \\
&\quad \times \exp\left(i\hat{n}(\theta) \cdot \left(k_F(\mathbf{x}_1 - \mathbf{x}') + k(\mathbf{x}' - \mathbf{x}_2)\right) + I_{\sigma,\hat{n}(\theta)}[\phi](\tau_1, \mathbf{x}_1; \tau_2, \mathbf{x}')\right) \\
&= \int d\eta \int \frac{dk k d\theta}{(2\pi)^2} \frac{T \csc\left(\pi T \left(-\frac{i\eta}{v_F} + \tau_2 - \tau_1\right)\right)}{2v_F} \times \\
&\quad \times \exp\left(i(k - k_F)\eta + ik\hat{n}(\theta) \cdot (\mathbf{x}_1 - \mathbf{x}_2) + I_{\sigma,\hat{n}(\theta)}[\phi](\tau_1, \mathbf{x}_1; \tau_2, \mathbf{x}_1 + \eta\hat{n}(\theta))\right).
\end{aligned} \tag{30}$$

Here we parametrized $\mathbf{x}' = \mathbf{x}_1 + \eta\hat{n}(\theta) + v\hat{n}(\theta + \pi/2)$ and integrated over $v$.

We consider $k_F \gg v_F T, |x_2 - x_1|^{-1}$ and a field $\phi$ without momentum components of order $k_F$. The $\eta$ integral can be viewed as a Fourier transform to the variable $k - k_F$. If we assume $v_F |\tau_2 - \tau_1| \gg k_F^{-1}$ then the $\eta$ dependence of the rest of the integrand has no momentum components of order $k_F$ and this Fourier transform will be small unless $|k - k_F| \ll k_F$. We later comment on what happens for $v_F |\tau_2 - \tau_1| \sim k_F^{-1}$. For the leading behavior we can thus assume $k$ is of order $k_F$. We then see that the exponent $ik\hat{n}(\theta) \cdot (\mathbf{x}_1 - \mathbf{x}_2)$ makes the $\theta$ integral oscillate rapidly except at the two points where $\hat{n}$ is parallel or anti-parallel to $\mathbf{x}_{12} = \mathbf{x}_2 - \mathbf{x}_1$. We perform saddle-point approximations around these points:

$$\begin{aligned}
G_{\text{IR},\sigma}^{\text{saddle point}}[\phi](x_1, x_2) &= -\sum_{s=\pm 1} \int \frac{d\eta dk}{(2\pi)^{3/2}} \frac{T \csc\left(\pi T \left(\tau_1 - \tau_2 + \frac{i\eta}{v_F}\right)\right)}{2v_F} \sqrt{\frac{k}{|\mathbf{x}_{12}|}} \times \\
&\quad \times \exp\left(i(k - k_F)\eta + is\pi/4 - isk|\mathbf{x}_{12}| + I_{\sigma,s\hat{x}_{12}}[\phi](\tau_1, \mathbf{x}_1; \tau_2, \mathbf{x}_1 + \eta s\hat{x}_{12})\right).
\end{aligned} \tag{31}$$

We proceed with the $k$ integral. It formally diverges but comes from the Fourier transform so we treat it as such and use

$$\int_0^\infty dk e^{ikz} \sqrt{k} \to \frac{\sqrt{\pi}}{2(-iz)^{3/2}}, \tag{32}$$

for these integrals. We then have

$$\begin{aligned}
G_{\text{IR},\sigma}^{\text{saddle point}}[\phi](x_1, x_2) &= -\sum_{s=\pm 1} \int d\eta e^{-i\eta k_F} \frac{T \csc\left(\pi T \left(\tau_1 - \tau_2 + \frac{i\eta}{v_F}\right)\right)}{8\pi v_F \sqrt{2|\mathbf{x}_{12}|}(-i(\eta - s|\mathbf{x}_{12}|))^{3/2}} \times \\
&\quad \times \exp\left(is\pi/4 + I_{\sigma,s\hat{x}_{12}}[\phi](\tau_1, \mathbf{x}_1; \tau_2, \mathbf{x}_1 + \eta s\hat{x}_{12})\right).
\end{aligned} \tag{33}$$

The $\eta$ integral can be viewed as the high frequency limit of a Fourier transform to the variable $k_F$. For $|\tau_2 - \tau_1| \gg 1/k_F$, the high frequency part of the transformed function is dominated by

the singularities at $\eta = s|\mathbf{x}_{12}|$. We can expand around them and perform the Fourier transform to get the leading large $k_F$ limit:

$$G_{\text{IR},\sigma}^{\text{saddle point}}[\phi](x_1, x_2) = -T\sqrt{\frac{k_F}{2\pi|\mathbf{x}_{12}|}}\sum_{s=\pm 1}\frac{e^{is\pi/4 - isk_F|\mathbf{x}_{12}| + I_{\sigma,s\hat{x}_{12}}[\phi](x_1, x_2)}}{2v_F\sin\left(\pi T\left(\tau_1 - \tau_2 + \frac{is|\mathbf{x}_{12}|}{v_F}\right)\right)} + \text{subleading}.$$

(34)

We would also like to consider equal-time correlation functions so now we consider the case where $v_F|\tau_2 - \tau_1| \sim k_F^{-1}$. Consider Eq. (30). The $\eta$ integral now gets a contribution also for $|k - k_F| \sim k_F$ since the fraction has frequencies of order $k_F$ when $\eta \sim v_F|\tau_2 - \tau_1|$. To calculate this contribution we make use of

$$I_{\sigma,\hat{n}}[\phi](x_1, x_2) = \mathcal{O}((\tau_2 - \tau_1)\partial_\tau \phi) + \mathcal{O}(|\mathbf{x}_2 - \mathbf{x}_1||\nabla\phi|).$$

(35)

Since $v_F|\tau_2 - \tau_1| \sim k_F^{-1}$ and $\phi$ contains no scales of order $k_F$ the first term can be neglected. The second term can also be neglected since $\eta \sim v_F|\tau_2 - \tau_1|$ for the contribution we are interested in. We can set $\phi = 0$ to leading order. Note that this is only true for the contribution missed in the saddle-point approximation. We thus have

$$G_{\text{IR}}[\phi](x_1, x_2) = G_{\text{IR}}^{\text{saddle point}}[\phi](x_1, x_2) + G_{\text{IR}}[0](x_1, x_2) - G_{\text{IR}}^{\text{saddle point}}[0](x_1, x_2) +$$
$$+ \text{subleading}.$$

(36)

Finding the free propagator in real space we find that the last two terms above cancel and the saddle point solution actually works for small $\tau_2 - \tau_1$ as well.

Now we consider two-point functions of the composite operators. Differentiating the generating functional with respect to the sources and only keeping the leading contribution for small $N_f$ we have:

$$\langle \rho_\sigma(0)\rho_\sigma(\tau, \mathbf{x})\rangle = -N_f\int \mathcal{D}\phi\, G_{\sigma,\text{IR}}[\phi](0;\tau,\mathbf{x})G_{\sigma,\text{IR}}[\phi](\tau,\mathbf{x};0)e^{-S_B[\phi]} + \text{subleading} \quad (37)$$

$$\langle b^\dagger(0)b(\tau,\mathbf{x})\rangle = N_f\int \mathcal{D}\phi\, G_{\uparrow,\text{IR}}[\phi](0;\tau,\mathbf{x})G_{\downarrow,\text{IR}}[\phi](0;\tau,\mathbf{x})e^{-S_B[\phi]} + \text{subleading}. \quad (38)$$

The determinant action has now been omitted since it is subleading in $N_f$. Note that the density correlator contains one more contraction but it is also subleading in $N_f$ and can be ommited, see Fig. 1. Note that from here on, in "subleading" we include terms that are either subleading in the small $N_f$ limit or in the $x \equiv |\mathbf{x}| \gg k_F^{-1}$ limit. We now integrate out the fields $\phi_a$. Expanding the background field Green's functions and combining terms we have:

$$\langle \rho_\sigma(0)\rho_\sigma(\tau,\mathbf{x})\rangle = -\frac{N_f k_F T^2}{8\pi v_F^2 x}\int \mathcal{D}\phi \sum_{\substack{s_1=\pm 1 \\ s_2=\pm 1}} \times$$

$$\times \frac{e^{i(s_1 - s_2)(\pi/4 - k_F x) + I_{\sigma,s_1\hat{x}}[\phi](0,0;\tau,\mathbf{x}) + I_{\sigma,s_2\hat{x}}[\phi](\tau,\mathbf{x};0,0)}}{\sin\left(\pi T\left(-\tau + \frac{is_1 x}{v_F}\right)\right)\sin\left(\pi T\left(\tau - \frac{is_2 x}{v_F}\right)\right)}e^{-S_B[\phi]} + \text{subleading} \quad (39)$$

$$\langle b^\dagger(0)b(\tau,\mathbf{x})\rangle = \frac{N_f k_F T^2}{8\pi v_F^2 x}\int \mathcal{D}\phi \sum_{\substack{s_1=\pm 1 \\ s_2=\pm 1}} \times$$

$$\times \frac{e^{i(s_1 + s_2)(\pi/4 - k_F x) + I_{\uparrow,s_1\hat{x}}[\phi](0,0;\tau,\mathbf{x}) + I_{\downarrow,s_2\hat{x}}[\phi](0,0;\tau,\mathbf{x})}}{\sin\left(\pi T\left(-\tau + \frac{is_1 x}{v_F}\right)\right)\sin\left(\pi T\left(-\tau + \frac{is_2 x}{v_F}\right)\right)}e^{-S_B[\phi]} + \text{subleading}, \quad (40)$$

where $\mathbf{x} \equiv x\hat{x}$. We have redefined $s_2 \to -s_2$ in (39) compared to (34) so that the four different $s_1, s_2$ contributions correspond to processes where the two fermions being exchanged between the bilinears live in the Fermi surface patches near $s_1 k_F \hat{x}$ and $s_2 k_F \hat{x}$, respectively. The $I_{\sigma,\hat{n}}[\phi]$ functions are linear in $\phi$ and should be treated as sources for $\phi$ in the path integral. We expand the two $I_{\sigma,\hat{n}}[\phi]$ functions and identify the sources:

$$I_{\sigma,s_1\hat{x}}[\phi](0,X) + I_{\sigma,s_2\hat{x}}[\phi](X,0) = \int d^3 X' \phi(X') \Big[ \lambda_{\sigma,s_1\hat{x}} f^+_{s_1\hat{x}}(X') - \lambda_{\sigma,s_1\hat{x}} f^+_{s_1\hat{x}}(X'-X) +$$

$$+ \lambda_{\sigma,s_2\hat{x}} f^+_{s_2\hat{x}}(X'-X) - \lambda_{\sigma,s_2\hat{x}} f^+_{s_2\hat{x}}(X') \Big]$$

$$\equiv - \int d^3 X' J^{\rho\rho}_{s_1,s_2,\lambda,X}(X') \phi(X') \tag{41}$$

$$I_{\uparrow,s_1\hat{x}}[\phi](0,X) + I_{\downarrow,s_2\hat{x}}[\phi](0,X) = \int d^3 X' \phi(X') \Big[ \lambda_{\uparrow,s_1\hat{x}} f^+_{s_1\hat{x}}(X') - \lambda_{\uparrow,s_1\hat{x}} f^+_{s_1\hat{n}}(X'-X) +$$

$$+ \lambda_{\downarrow,s_2\hat{x}} f^+_{s_2\hat{x}}(X') - \lambda_{\downarrow,s_2\hat{x}} f^+_{s_2\hat{x}}(X'-X) \Big]$$

$$\equiv - \int d^3 X' J^{b^\dagger b}_{s_1,s_2,\lambda,X}(X') \phi(X'). \tag{42}$$

Here we use $X = (\tau, \mathbf{x})$ to avoid confusion with $x = |\mathbf{x}|$. $S_B[\phi]$ is diagonal in momentum space so we write the source terms in momentum space:

$$\int d^3 X' J^{\rho\rho}_{s_1,s_2,\lambda,X}(X') \phi(X') = T \sum_{\omega_n} \int \frac{d^2\mathbf{k}}{(2\pi)^2} J^{\rho\rho}_{s_1,s_2,\lambda,X}(k) \phi(-k) \tag{43}$$

$$\int d^3 X' J^{b^\dagger b}_{s_1,s_2,\lambda,X}(X') \phi(X') = T \sum_{\omega_n} \int \frac{d^2\mathbf{k}}{(2\pi)^2} J^{b^\dagger b}_{s_1,s_2,\lambda,X}(k) \phi(-k), \tag{44}$$

where

$$J^{\rho\rho}_{s_1,s_2,\lambda,X}(k) = \Big( \lambda_{\sigma,s_1\hat{x}} f^+_{s_1\hat{x}}(k) - \lambda_{\sigma,s_2\hat{x}} f^+_{s_2\hat{x}}(k) \Big) \Big( e^{-iXk} - 1 \Big) \tag{45}$$

$$J^{b^\dagger b}_{s_1,s_2,\lambda,X}(k) = \Big( \lambda_{\uparrow,s_1\hat{x}} f^+_{s_1\hat{x}}(k) + \lambda_{\downarrow,s_2\hat{x}} f^+_{s_2\hat{x}}(k) \Big) \Big( e^{-iXk} - 1 \Big) \tag{46}$$

$$f^+_{\hat{n}}(k) = \frac{1}{i\omega_n - v_F \hat{n} \cdot \mathbf{k}}, \tag{47}$$

and the sum is over bosonic Matsubara frequencies $\omega_n = 2\pi n T$. Now we perform the path integral over $\phi$ for a general source $J(k)$ and the bosonic action $S_B[\phi]$:

$$\int \mathcal{D}\phi \exp\left( T \sum_{\omega_n} \int \frac{d^2\mathbf{k}}{(2\pi)^2} \Big[ -J(k)\phi(-k) - \frac{\phi(k)\phi(-k)}{2D(k)} \Big] \right) =$$

$$= \exp\left( T \sum_{\omega_n} \int \frac{d^2\mathbf{k}}{(2\pi)^2} \frac{J(k)J(-k)D(k)}{2} \right), \tag{48}$$

where

$$D(k) = \frac{1}{\omega_n^2 + c^2 k_x^2 + c^2 k_y^2 + r} \, . \tag{49}$$

Now consider the exponent of the RHS of Eq. (48) with the source $J^{\rho\rho}_{s_1,s_2,\lambda,X}(k)$ and $J^{b^\dagger b}_{s_1,s_2,\lambda,X}(k)$:

$$
\begin{aligned}
E^{\rho\rho}_{s_1,s_2,\lambda}(X) &\equiv T \sum_{\omega_n} \int \frac{d^2\mathbf{k}}{(2\pi)^2} \frac{J^{\rho\rho}_{s_1,s_2,\lambda,X}(k) J^{\rho\rho}_{s_1,s_2,\lambda,X}(-k) D(k)}{2} \\
&= T \sum_{\omega_n} \int \frac{d^2\mathbf{k}}{(2\pi)^2} \left( \lambda_{\sigma,s_1\hat{x}} f^+_{s_1\hat{x}}(k) - \lambda_{\sigma,s_2\hat{x}} f^+_{s_2\hat{x}}(k) \right) \times \\
&\qquad \times \left( \lambda_{\sigma,s_1\hat{x}} f^+_{s_1\hat{x}}(-k) - \lambda_{\sigma,s_2\hat{x}} f^+_{s_2\hat{x}}(-k) \right) (1 - \cos(Xk)) D(k) \\
&= -\lambda^2_{\sigma,s_1\hat{x}} h_{s_1\hat{x},s_1\hat{x}}(X) + 2\lambda_{\sigma,s_1\hat{x}} \lambda_{\sigma,s_2\hat{x}} h_{s_1\hat{x},s_2\hat{x}}(X) - \lambda^2_{\sigma,s_2\hat{x}} h_{s_2\hat{x},s_2\hat{x}}(X)
\end{aligned}
\tag{50}
$$

$$
\begin{aligned}
E^{b^\dagger b}_{s_1,s_2,\lambda}(X) &\equiv T \sum_{\omega_n} \int \frac{d^2\mathbf{k}}{(2\pi)^2} \frac{J^{b^\dagger b}_{s_1,s_2,\lambda,X}(k) J^{b^\dagger b}_{s_1,s_2,\lambda,X}(-k) D(k)}{2} \\
&= T \sum_{\omega_n} \int \frac{d^2\mathbf{k}}{(2\pi)^2} \left( \lambda_{\uparrow,s_1\hat{x}} f^+_{s_1\hat{x}}(k) + \lambda_{\downarrow,s_2\hat{x}} f^+_{s_2\hat{x}}(k) \right) \times \\
&\qquad \times \left( \lambda_{\uparrow,s_1\hat{x}} f^+_{s_1\hat{x}}(-k) + \lambda_{\downarrow,s_2\hat{x}} f^+_{s_2\hat{x}}(-k) \right) (1 - \cos(Xk)) D(k) \\
&= -\lambda^2_{\uparrow,s_1\hat{x}} h_{s_1\hat{x},s_1\hat{x}}(X) - 2\lambda_{\uparrow,s_1\hat{x}} \lambda_{\downarrow,s_2\hat{x}} h_{s_1\hat{x},s_2\hat{x}}(X) - \lambda^2_{\downarrow,s_2\hat{x}} h_{s_2\hat{x},s_2\hat{x}}(X),
\end{aligned}
\tag{51}
$$

where

$$h_{\hat{n}_1,\hat{n}_2}(X) \equiv T \sum_n \int \frac{d^2\mathbf{k}}{(2\pi)^2} (\cos(Xk) - 1) f^+_{\hat{n}_1}(k) f^+_{\hat{n}_2}(-k) D(k). \tag{52}$$

We can now use this for the path integrals in Eq. (39) and Eq. (40):

$$\langle \rho_\sigma(0)\rho_\sigma(\tau,\mathbf{x}) \rangle = \frac{N_f k_F T^2}{8\pi v_F^2 x} \sum_{\substack{s_1=\pm 1 \\ s_2=\pm 1}} \frac{\exp\left( i(s_1 - s_2)(\pi/4 - k_F x) + E^{\rho\rho}_{s_1,s_2,\lambda}(X) \right)}{\sin\left( \pi T \left( -\tau + \frac{is_1 x}{v_F} \right) \right) \sin\left( \pi T \left( -\tau + \frac{is_2 x}{v_F} \right) \right)} + \text{sub.} \tag{53}$$

$$\langle b^\dagger(0)b(\tau,\mathbf{x}) \rangle = \frac{N_f k_F T^2}{8\pi v_F^2 x} \sum_{\substack{s_1=\pm 1 \\ s_2=\pm 1}} \frac{\exp\left( i(s_1 + s_2)(\pi/4 - k_F x) + E^{b^\dagger b}_{s_1,s_2,\lambda}(X) \right)}{\sin\left( \pi T \left( -\tau + \frac{is_1 x}{v_F} \right) \right) \sin\left( \pi T \left( -\tau + \frac{is_2 x}{v_F} \right) \right)} + \text{sub.} \tag{54}$$

The $E_{s_1,s_2,\lambda}(X)$-functions should be understood as capturing the interaction corrections to the contribution to the real space fermion four-point function where one fermion is in the patch at $s_1 k_F \hat{x}$ and the other is in the patch at $s_2 k_F \hat{x}$.

Furthermore, these functions are composed of the $h_{\hat{n}_1,\hat{n}_2}(X)$-functions which capture the effects of a fermion in patch $\hat{n}_1 k_F$ exchanging a boson with a fermion in patch $\hat{n}_2 k_F$. Note that the $E_{s_1,s_2,\lambda}(X)$ functions contain $h_{s_1\hat{x},s_1\hat{x}}$ even when $s_1 \neq s_2$. These are the self-energy corrections, the fermions always exchange bosons with themselves and thus within the same patch. Also note that $E^{\rho\rho}_{s,s,\lambda,X} = 0$, the self-energy corrections precisely cancel with the processes that exchange bosons between the two fermions for the $\langle \rho\rho \rangle$ correlator as expected from [36].

We define

$$h^\pm(\tau, x) = T \sum_n \int \frac{dk_x dk_y}{(2\pi)^2} \frac{\cos(\omega_n \tau - k_x x) - 1}{(i\omega_n - v_F k_x)(-i\omega_n \pm v_F k_x)} D(\omega_n, k_x, k_y),  \qquad (55)$$

and note that

$$h_{\hat{x},\hat{x}}(X) = h^+(\tau, x) \qquad (56)$$

$$h_{-\hat{x},\hat{x}}(X) = h^-(\tau, x) \qquad (57)$$

$$h_{\hat{x},-\hat{x}}(X) = h^-(\tau, x) \qquad (58)$$

$$h_{-\hat{x},-\hat{x}}(X) = h^+(\tau, -x). \qquad (59)$$

We additionally define $\lambda_\sigma^\pm = \lambda_{\sigma, \pm\hat{x}}$ and expand the $s_1, s_2$ summations to obtain:

$$\langle \rho_\sigma(0,0)\rho_\sigma(\tau, \mathbf{x})\rangle = \frac{N_f k_F T^2}{8\pi v_F^2 x}\Bigg[ \csc^2\left(\pi T(\tau - \frac{ix}{v_F})\right) + \csc^2\left(\pi T(\tau + \frac{ix}{v_F})\right)$$

$$+ \frac{4\sin(2k_F x)\exp\left(\sum_a \left[2\lambda_{\sigma a}^- \lambda_{\sigma a}^+ h_a^-(\tau, x) - \lambda_{\sigma a}^{-2} h_a^+(\tau, -x) - \lambda_{\sigma a}^{+2} h_a^+(\tau, x)\right]\right)}{\cosh\left(\frac{2\pi T x}{v_F}\right) - \cos(2\pi\tau T)}\Bigg]$$

$$+ \text{subleading} \qquad (60)$$

$$\langle b^\dagger(0)b(\tau, \mathbf{x})\rangle = \frac{N_f k_F T^2}{8\pi v_F^2 x}\Bigg( \csc\left(\pi T\left(\tau - \frac{ix}{v_F}\right)\right) \csc\left(\pi T\left(\tau + \frac{ix}{v_F}\right)\right)$$

$$\times \Bigg[ \exp\left(-\sum_a \left[\lambda_{\uparrow a}^{+2} h_a^+(\tau, x) + 2\lambda_{\downarrow a}^- \lambda_{\uparrow a}^+ h_a^-(\tau, x) + \lambda_{\downarrow a}^{-2} h^+(\tau, -x)\right]\right)$$

$$+ \exp\left(-\sum_a \left[\lambda_{\downarrow a}^{+2} h_a^+(\tau, x) + 2\lambda_{\uparrow a}^- \lambda_{\downarrow a}^+ h^-(\tau, x) + \lambda_{\uparrow a}^{-2} h_a^+(\tau, -x)\right]\right)\Bigg]$$

$$+ i\csc^2\left(\pi T\left(\tau - \frac{ix}{v_F}\right)\right) \exp\left(-2ik_F x - \sum_a (\lambda_{\downarrow a}^- + \lambda_{\uparrow a}^-)^2 h_a^+(\tau, -x)\right)$$

$$- i\csc^2\left(\pi T\left(\tau + \frac{ix}{v_F}\right)\right) \exp\left(2ik_F x - \sum_a (\lambda_{\downarrow a}^+ + \lambda_{\uparrow a}^+)^2 h_a^+(\tau, x)\right)\Bigg)$$

$$+ \text{subleading}. \qquad (61)$$

Here we have reintroduced the $a$ indices and the $h_a^\pm(\tau, x)$ are defined with the propagator $D_a(\omega_n, k_x, k_y)$ for the field $\phi_a$.

## 3 Divergences in long distance correlation functions

At the critical points $T = 0$, $r_a = 0$ there is no internal scale in (55) so by dimensional analysis we have

$$h_a^\pm(\tau = 0, x) \sim x. \qquad (62)$$

For $\tau = 0$, the $h_a^\pm$ functions are linear in the separation $x$ and the prefactors of $x$ in the exponents of Eq. (60) and Eq. (61) are quadratic forms of the different coupling constants $\lambda_{\uparrow,\downarrow}^\pm$. It is possible to find couplings such that correlations grow exponentially in the spatial

separation $x$ unless these quadratic forms are negative semi-definite. An unbounded growth is unphysical so it is interesting to investigate whether it appears for any setup.

In the introduction we mentioned that quantum critical fluctuations might induce instabilities. A treatment of a theory with an instability by expansion around the naive vacuum where the expectation values of all fields of Eq. (1) are 0 will then be incorrect. Excitations should instead be considered from the true vacuum, which breaks some of the symmetries of the action. We do not attempt to describe the true ground state but instead study expectation values obtained by expansion around the naive vacuum. Although these expectation values can not be trusted if an instability is present, they can show inconsistencies, like the above unbounded growth, that indicate the presence of an instability. Specifically, we study how the correlation functions we calculated in the previous section behaves at large spatial separations. The correlators are expected to decay at a certain rate with separation and when this is not the case, we attribute this to an instability.

With that as motivation, we study the behaviors of these exponents in the long distance limit in this section. We do that at the QCP and away from it, both at $T = 0$ and $T > 0$. We start out by only considering fermions coupled to one order parameter fluctuation field so we drop the index $a$ for now.

Before calculating the $h^\pm$ functions in different limits we give a physical interpretation of them and find some of their general properties. Looking back at where these functions show up we see that $h^+$ captures processes where the fermion exchanges bosons with fermions within the same patch of the Fermi surface. The fermion self-energy is obtained from this type of process so quasi-particle smearing will be governed by the behavior of $h^+$. $h^-$ captures processes where a fermion exchanges bosons with a fermion in an opposite patch of the Fermi surface. This can give rise to the attractive glue needed for e.g. pairing.

Using that $D(\omega, k_x, k_y)$ is positive and even in energy and momentum we have

$$h^\pm(-\tau, -x) = h^\pm(\tau, x) \tag{63}$$

$$h^+(-\tau, x) = \overline{h^+(\tau, x)} \tag{64}$$

$$h^-(-\tau, x) = h^-(\tau, x) \tag{65}$$

$$h^-(\tau, x) \leq 0 \tag{66}$$

$$|h^+(\tau, x)| \leq |h^-(\tau, x)|, \tag{67}$$

where we used the triangle inequality to move the absolute value inside the integral of Eq. (55) for the last inequality. This last identity will turn out to be important, it means that the interactions between opposing patches in a sense is stronger than the effects of quasi-particle smearing.

## 3.1 The quantum critical point

We now consider the theory at the quantum critical point where $T = r = 0$. We can write the correlation functions as

$$\langle \rho_\sigma(0,0)\rho_\sigma(\tau, \mathbf{x}) \rangle = \frac{N_f k_F}{4\pi^3 x} \left[ \frac{\sin(2k_F x)\exp\left(\boldsymbol{\lambda}_\sigma^T A \boldsymbol{\lambda}_\sigma\right)}{v_F^2 \tau^2 + x^2} + \frac{v_F^2 \tau^2 - x^2}{(v_F^2 \tau^2 + x^2)^2} \right] \tag{68}$$

$$\langle b^\dagger(0)b(\tau, \mathbf{x}) \rangle = \frac{N_f k_F}{8\pi^3 x} \left[ \frac{\exp\left(\boldsymbol{\lambda}^{+T} B \boldsymbol{\lambda}^+\right) + \exp\left(\boldsymbol{\lambda}^{-T} B \boldsymbol{\lambda}^-\right)}{v_F^2 \tau^2 + x^2} \right.$$
$$- i \frac{1}{(x + iv_F \tau)^2} \exp\left(-2ik_F x - (\lambda_\downarrow^- + \lambda_\uparrow^-)^2 h^+(\tau, -x)\right)$$
$$\left. + i \frac{1}{(x - iv_F \tau)^2} \exp\left(2ik_F x - (\lambda_\downarrow^+ + \lambda_\uparrow^+)^2 h^+(\tau, x)\right) \right], \tag{69}$$

where

$$A = \begin{bmatrix} -h^+(\tau,x) & h^-(\tau,x) \\ h^-(\tau,x) & -\overline{h^+(\tau,x)} \end{bmatrix}, \quad \boldsymbol{\lambda}_\sigma = \begin{bmatrix} \lambda_\sigma^+ \\ \lambda_\sigma^- \end{bmatrix} \tag{70}$$

$$B = \begin{bmatrix} -h^+(\tau,x) & -h^-(\tau,x) \\ -h^-(\tau,x) & -\overline{h^+(\tau,x)} \end{bmatrix}, \quad \boldsymbol{\lambda}^\pm = \begin{bmatrix} \lambda_\uparrow^\pm \\ \lambda_\downarrow^\mp \end{bmatrix}. \tag{71}$$

We diagonalize $A, B$ to find out whether they are positive-/negative- or indefinite. The eigenvalues of the above matrices are both are given by

$$A_{1,2} = B_{1,2} = \pm\sqrt{h^-(\tau,x)^2 - \mathrm{Im}(h^+(\tau,x))^2} - \mathrm{Re}(h^+(\tau,x)). \tag{72}$$

By using Eq. (67) we find that

$$A_1 = B_1 \geq 0 \tag{73}$$
$$A_2 = B_2 \leq 0. \tag{74}$$

The exponents are thus indefinite quadratic forms of the couplings, regardless of the form of $D(\omega,\mathbf{k})$. This means that it is, for both correlators, possible to find couplings $\lambda_\sigma^\pm$ such that the exponent is positive, regardless of the form of $D(\omega,\mathbf{k})$. For a scale-free $D(\omega,\mathbf{k})$ at the critical point we can find the dependence of the separation by a simple scaling argument. Instead of considering the uncorrected Green's function of Eq. (49), we might as well consider a generalization with critical exponent $\eta$:

$$D(\omega,k_x,k_y) = \frac{1}{(\omega^2 + c^2 k_x^2 + c^2 k_y^2)^{(3-\eta)/2}}. \tag{75}$$

By rescaling the momentum integrals we have that

$$h^\pm(s\tau,sx) = s^{2-\eta}h^\pm(\tau,x). \tag{76}$$

We find that there are couplings such that the exponents of Eq. (60) and Eq. (61) grow unboundedly in the separation for all critical exponents $\eta < 2$. This leads to unphysical correlation functions that similarly grow indefinitely in the separation. In the case of the uncorrected Green's function for $\phi$ in our action we have $\eta = 1$. This unbounded growth of correlations is interpreted as an instability of the theory. We discuss this in more detail in subsection 3.3 and in section 5 where we consider a type of QCP at a time.

Now we investigate for what couplings we find instabilities. Considering only a spatial separation, we have

$$|\lambda_\sigma^+ + \lambda_\sigma^-| < R|\lambda_\sigma^+ - \lambda_\sigma^-| \implies \text{unbounded } \langle\rho_\sigma\rho_\sigma(x)\rangle \text{ growth}$$
$$|\lambda_\uparrow^+ - \lambda_\downarrow^-| < R|\lambda_\uparrow^+ + \lambda_\downarrow^-| \implies \text{unbounded } \langle b^\dagger b(x)\rangle \text{ growth}$$
$$|\lambda_\downarrow^+ - \lambda_\uparrow^-| < R|\lambda_\downarrow^+ + \lambda_\uparrow^-| \implies \text{unbounded } \langle b^\dagger b(x)\rangle \text{ growth}, \tag{77}$$

where

$$0 < R \equiv \sqrt{\frac{h^-(0,x) + h^+(0,x)}{h^-(0,x) - h^+(0,x)}}. \tag{78}$$

Note that $R$ is independent of $x$ at the QCP.

We have not yet considered the oscillating part of the pair-pair correlation function. To do this we need to calculate $h_\sigma^+$ to find its sign. We do this for a spatial separation $x$ and find that it is positive everywhere, see Fig. 2 (result presented in Appendix A). Since it is positive we find that the last two terms of Eq. (69) decay for all couplings. We only considered spatial separations here but for $\tau \neq 0$ it is possible to obtain an $h^+(\tau,x)$ that grows in $\tau$. However, this is harder to interpret since $\tau$ is Euclidean time. It seems likely that when continued to real time this once again is negative but that is something left for the future to be studied.

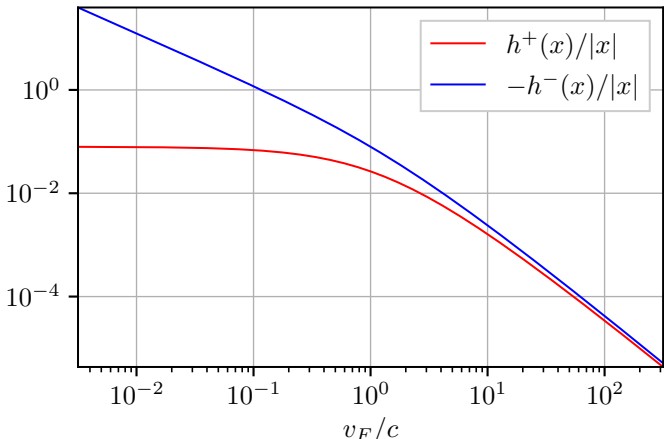

Figure 2: The $h^{\pm}(\tau = 0, x)$ are linear in $x$ at the critical point using the uncorrected $\phi$ Green's function of Eq. (49). This figure shows the prefactor of $x$ for different ratios of the Fermi velocity to the $\phi$ velocity $c$.

## 3.2 Zero temperature - away from criticality

We now continue to work at $T = 0$ but away from the critical point. We consider $D(\omega, \mathbf{k})$ of Eq. (49) with a finite gap $r > 0$ and once again calculate the above considered correlation functions at large spatial separations to see whether they show signs of instabilities. We have additional symmetry at the point where $c = v_F$ (see Chapter 5 of [37]) and we can calculate $h^{\pm}$ explicitly there. We find

$$h^+(0, x) = \frac{2v_F \exp\left(-\frac{\sqrt{r}|x|}{v_F}\right)(v_F + \sqrt{r}|x|) + rx^2 - 2v_F^2}{8\pi v_F^2 r^{3/2} x^2} \tag{79}$$

$$h^-(0, x) = \frac{\text{Ei}\left(-\frac{\sqrt{r}|x|}{v_F}\right) - \log\left(\frac{\sqrt{r}|x|}{v_F}\right) - \gamma}{4\pi v_F^2 \sqrt{r}}, \tag{80}$$

where Ei is the exponential integral and $\gamma$ is Euler's constant. Although not immediately evident from these expressions, for small $x$ this behaves as in the critical case and $h^{\pm}$ are linear in $|x|$. This linear growth slows down at a separation $|x| \sim v_F/\sqrt{r}$ and in the case of $h^+$ it approaches a constant while for $h^-$ it crosses over to a logarithmic growth. These asymptotic behaviors have been calculated for a general $v_F/c$ (see Appendix B):

$$h^+(\tau, x) = \text{finite as } x \to \infty \tag{81}$$

$$h^-(\tau, x) = -\frac{\log\left(r\left(\frac{x^2}{v_F^2} + \tau^2\right)\right)}{8\pi c \sqrt{r} v_F} + \text{finite as } x \to \infty. \tag{82}$$

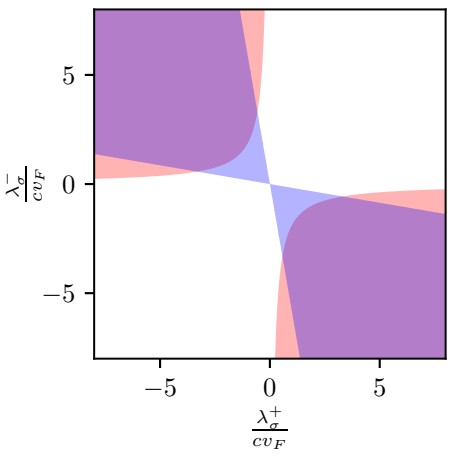

(a) Unbounded growth in $\langle \rho_\sigma \rho_\sigma \rangle$

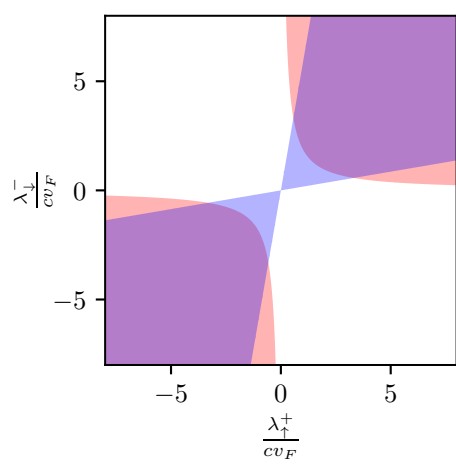

(b) Unbounded growth in $\langle b^\dagger b \rangle$

Figure 3: The colored regions indicate couplings for which the corresponding correlator grows unboundedly in the separation at $T = 0$. The plots show this both at the QCP (blue) and for $\sqrt{r} = 0.1$ (red). Note that the red regions move closer to the origin as $r \to 0$ and thus cover the whole quadrants in the critical limit. Here we have used $c = v_F$ and thus $R = 1/\sqrt{2}$.

The $h^-$ function will thus dominate the exponents of Eq. (60) and (61) for large separations. This gives corrections to the power-law of the free theory, $x^{-3}$, at long distances:

$$\langle \rho_\sigma \rho_\sigma(\tau, x) \rangle = C_1 N_f k_F \frac{\sin(2k_F x)}{x} \left( v_F^2 \tau^2 + x^2 \right)^{-\frac{\lambda_\sigma^- \lambda_\sigma^+}{4\pi c v_F \sqrt{r}} - 1} +$$
$$+ N_f k_F \frac{v_F^2 \tau^2 - x^2}{4\pi^3 x (v_F^2 \tau^2 + x^2)^2} + \text{subleading} \tag{83}$$

$$\langle b^\dagger b(\tau, x) \rangle = C_2 N_f k_F x^{-1} \left( v_F^2 \tau^2 + x^2 \right)^{\frac{\lambda_\downarrow^- \lambda_\uparrow^+}{4\pi c v_F \sqrt{r}} - 1} +$$
$$+ C_3 N_f k_F x^{-1} \left( v_F^2 \tau^2 + x^2 \right)^{\frac{\lambda_\uparrow^- \lambda_\downarrow^+}{4\pi c v_F \sqrt{r}} - 1} +$$
$$- iC_4 \frac{N_f k_F \exp(-2ik_F x)}{x(x + iv_F \tau)^2} + iC_5 \frac{N_f k_F \exp(2ik_F x)}{x(x - iv_F \tau)^2} + \text{subleading}. \tag{84}$$

The prefactors $C_i$ are given by the asymptotic value of $h^+$ and the finite part of $h^-$ and depend on $\lambda_\sigma^{\pm,2}, v_F, c, r$. Note that these expressions apply to any gapped boson propagator under the constraints mentioned in Appendix B. Interestingly, the corrected exponents may become larger than $1/2$ meaning that the correlation functions grow unboundedly for large separations, just as at the QCP. The growth is now a power-law with an exponent that approaches $\infty$ as we approach the QCP where $r = 0$. The conditions for unbounded growth are:

$$\lambda_\sigma^- \lambda_\sigma^+ < -6\pi c v_F \sqrt{r} \implies \text{unbounded } \langle \rho_\sigma \rho_\sigma(x) \rangle \text{ growth} \tag{85}$$
$$\lambda_\downarrow^- \lambda_\uparrow^+ \text{ or } \lambda_\uparrow^- \lambda_\downarrow^+ > 6\pi c v_F \sqrt{r} \implies \text{unbounded } \langle pp(x) \rangle \text{ growth}. \tag{86}$$

These are not the same conditions as what we had at the QCP in the limit $r \to 0$, see Figure 3. The difference can be understood as the $|\mathbf{x}| \to \infty$ and $r \to 0$ limits not commuting. The $h^+$ function becomes important when taking $r \to 0$ first and affects the large $|\mathbf{x}|$ behavior. However, it is neglected when considering $|\mathbf{x}| \to \infty$ at any finite $r$ since $h^+$ approaches a constant (that diverges as $r \to 0$).

The correlation functions do not have to grow unboundedly to indicate instabilities. The static pair and density susceptibilities can be calculated through a Fourier transform of the real space correlation functions where we neglect small separations and only consider $x > \Lambda, \Lambda \gg 1/k_F$. The static susceptibilities give the linear respons to sources for a pair or density operator. A diverging static susceptibility indicates a finite response without a source and thus an instability. First let us consider the $T = 0$ static susceptibilites of the free theory. Performing the $\tau$ and angular integral we have

$$\chi_{\rho,0}(\mathbf{q}) = \int d\tau d^2 x e^{i\mathbf{q}\cdot\mathbf{x}}\langle \rho(0)\rho(\tau,\mathbf{x})\rangle = \frac{N_f k_F}{2\pi v_F}\int_\Lambda^\infty dx \frac{J_0(qx)\sin(2k_F x)}{x} + \text{finite} \tag{87}$$

$$\chi_{b,0}(\mathbf{q}) = \int d\tau d^2 x e^{i\mathbf{q}\cdot\mathbf{x}}\langle b^\dagger(0)b(\tau,\mathbf{x})\rangle = \frac{N_f k_F}{4\pi v_F}\int_\Lambda^\infty dx \frac{J_0(qx)}{x} + \text{finite}, \tag{88}$$

where $J_0$ is the zeroth Bessel function of the first kind. The density susceptibility is finite for all $q$ whereas the pair susceptibility diverges at $q = 0$. There is an instability at $T = 0$ in the free theory leading to BCS superconductivity when we add any finite attractive four-Fermi interaction. Let us now see how interactions in the $N_f \to 0$ limit change this. We only consider the large $x$ asymptotics of the integral to see whether it diverges in the IR so we can use Eq. (83)–(84). We find that the density susceptibility diverges at $q = 2k_F$ when

$$\sqrt{r} < \sqrt{r_c} \equiv -\frac{\lambda_\sigma^- \lambda_\sigma^+}{\pi c v_F}. \tag{89}$$

This means that whenever $\lambda_\sigma^- \lambda_\sigma^+ < 0$, there is a critical gap $0 < r < r_c$ for the field $\phi$ where an instability is seen in the density correlator at wavevector $Q = 2k_F$. Considering the pair-susceptibility, we find that the instability of the free theory is cured when both of these two conditions are satisfied:

$$\lambda_\uparrow^- \lambda_\downarrow^+ < 0 \tag{90}$$

$$\lambda_\downarrow^- \lambda_\uparrow^+ < 0. \tag{91}$$

We have a faster divergence in the opposite case where the left hand sides are positive and the integral diverges algebraically instead of logarithmically in the IR.

## 3.3 Spontaneous symmetry breaking

In the previous subsections we found combinations of parameters where correlations of operators at different points grow unboundedly in their separation, and in a broader region, diverging static susceptibilities. We interpret the divergences as instabilities towards states where the corresponding operators have infinite expectation values. In practice, this is cut off by higher order terms that we have omitted in the action, but which become important for large values of the fields. The true ground states of the physical theories we are interested in are expected to instead be states where these operators have finite expectation values. We can not find the exact ground states without considering these higher order terms. The typical approach is to use mean-field theory to study the ground state. We leave that for future work and instead make the following natural guesses:

$$\langle b(x)\rangle = A\exp^{i\theta} \quad \text{for pairing susceptibility divergence} \tag{92}$$

$$\langle \rho_\sigma(x)\rangle \sim B\sin(2k_F \mathbf{x}\cdot\hat{n} + \theta) \quad \text{for charge/spin susceptibility divergence}, \tag{93}$$

where $0 < A, 0 < \theta < 2\pi$. The first is a pairing state that spontaneously breaks the $U(N_f)$ symmetry whereas the second is a charge or spin order that breaks translation symmetry. The

form of the density fluctuation is only symbolic, we expect oscillations around the momentum $\sim 2k_F$ but how this happens in practice depends on the angular dependence of the coupling and what higher order terms stop the infinite growth of the correlator. We consider the expected ground states in more detail for specific systems in Section 5.

## 3.4 Finite temperature

The unbounded growths of correlations at $T = 0$ are interpreted as instabilities towards symmetry-breaking ground state. However, spontaneous symmetry breaking of continuous symmetries is not allowed at finite temperature in two dimensions for short range interactions by the Hohenberg-Mermin-Wagner [38] theorem. The interactions mediated by $\phi$ are short range for any finite $r$ so we do not expect correlations to grow in the separation at finite temperatures away from $r = 0$. Fluctuations in the parameter $\theta$ in Eq. (92) and Eq. (93) are the Goldstone modes and these proliferate at finite temperature destroying any long-range order. However, it could still be possible to find quasi-long-range order such as in the two-dimensional XY model below the Berezinskii-Kosterlitz-Thouless transition [39]. A phase with quasi-long-range order has correlators that decay with a power-law of the separation at finite temperature.

Now we consider the long distance behavior of the pair and density correlation functions at a finite temperature $T$ to verify lack of spontaneous symmetry breaking and to see whether they exhibit quasi-long-range order. The $h^\pm$ functions are calculated through a Matsubara sum at finite temperature. It is useful to consider the zeroth Matsubara frequency ($n = 0$) and the rest of the sum ($n \neq 0$) separately. This is also done for the self-energy in [35, 40, 41] where these contributions are labeled NFL/quantum and thermal/classical, respectively. We define:

$$h^\pm(\tau, x) = h_{n=0}^\pm(x) + h_{n\neq 0}^\pm(\tau, x). \tag{94}$$

Note that the functions $h_{n=0}^\pm$ and $h_{n\neq 0}^\pm$ individually satisfy the properties in Eq. (63) to (67) and that

$$h_{n=0}^+(x) = -h_{n=0}^-(x). \tag{95}$$

Additionally, using

$$0 \leq \int \frac{dk_y}{2\pi} D(\omega_n, k_x, k_y) \leq \frac{1}{2c|\omega_n|}, \tag{96}$$

we find that

$$|h_{n\neq 0}^\pm(\tau, x)| \leq \frac{1}{24 v_F c T}. \tag{97}$$

The $n \neq 0$ contributions will thus not be very important since they are finite for large $x$. We can express the thermal $h_{n=0}^\pm(x)$ in terms of a Meijer $G$-function, see Appendix C. They diverge logarithmically for small $r$:

$$h_{n=0}^\pm(\tau, x) = \mp \frac{Tx^2}{8\pi c^2 v_F^2} \log\left(\frac{rx^2}{c^2}\right) + \text{finite}. \tag{98}$$

This is an IR divergence. The same divergence shows up at each order in perturbation theory and can be seen by calculating the fermion self-energy perturbatively (see Figure 4):

$$\Sigma_{\lambda^2}(\omega_n, k_x) = \frac{T\lambda^2}{2\pi c} \sum_{\omega_m} \frac{\cos^{-1}\left(\frac{c}{v_F} \frac{ik_x v_F + \omega_m + \omega_n}{\sqrt{r + \omega_m^2}}\right)}{\sqrt{c^2(ik_x v_F + \omega_m + \omega_n)^2 - v_F^2\left(r + \omega_m^2\right)}}. \tag{99}$$

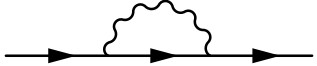

Figure 4: The single diagram contributing to the fermion self-energy at second order in the coupling.

The $\omega_m = 0$ contribution diverges logarithmically as $r \to 0$. Despite these IR divergences at each order, once all perturbative corrections are summed up into the exponentials of Eq. (60) and (61), we find that the correlation functions do not diverge in the $r \to 0$ limit. This can be seen by consider the matrices $A, B$ as in Eq. (68) and (69). Using Eq. (95) and Eq. (97) we can write

$$A = h_{n=0}^+(\tau, x) \begin{bmatrix} -1 & -1 \\ -1 & -1 \end{bmatrix} + \text{IR finite} \tag{100}$$

$$B = h_{n=0}^+(\tau, x) \begin{bmatrix} -1 & 1 \\ 1 & -1 \end{bmatrix} + \text{IR finite}. \tag{101}$$

Now both $A$ and $B$ are negative semidefinite (in the limit $r \to 0$) so, regardless of the couplings, the IR divergence does not make these correlators diverge but may suppresses components of them to 0. We have thus found that correlation functions diverge as we take $r \to 0$ at any finite order in perturbation theory. However, as we sum up all diagrams in the $N_f \to 0$ limit, we can safely take $r \to 0$ and the considered correlation functions remain finite. This is similar to what was found in [33]. They study the same model (albeit at matrix large $N$) but in $d = 3 - \epsilon$ dimensions. Similarly they find an IR divergence in the form of a diverging decay rate of electrons, in a perturbative treatment. The matrix large $N$ limit allows them to sum contributions at all orders and they find that this cures the IR divergence and they obtain a non-diverging fermion self-energy. A recent work has considered the same limit in $d = 2$ [41]. The IR divergence is then more severe and they find that it is not cured by summation of rainbow diagrams. Instead they argue that a $\phi^4$ interaction that is irrelevant at $T = 0$ becomes relevant at finite $T$ and there contributes a mass to the boson that cures the IR divergences. It is interesting that summation of rainbow diagrams did not cure the IR divergence while the $N_f \to 0$ sum of diagrams cured it without any corrections to the boson. Their non-cured IR-divergence manifests itself as a diverging fermion self-energy. We note that if we would calculate the fermion Green's function to leading order for $x \gg k_F^{-1}$ at finite $T$, this would be suppressed by $\exp(-\lambda^2 h^+)$ to identically 0 as $r \to 0$ and once Fourier transformed also result in a diverging self-energy. However, since the leading term in the large $xk_F$ expansion is suppressed, one has to include subleading terms (which are not necessarily suppressed to 0) before Fourier transforming and calculating the self-energy. This means that whether our fermion self-energy is finite or not depends on the Fermi surface curvature and cannot be found in the linearized patch we have used, but which is also used in [41]. So to conclude, up to the approximations we have both made, our results agree on a diverging Fermion self-energy, but we expect this to simply be caused by the linearized patch in our case and it does not manifest itself in the real space observables we consider.

Let us now depart from the critical, but finite temperature, theory and consider our correlators at a finite $r$ and $T$ and a large separation ($x \gg c/\sqrt{r}$ and $x \gg v_F/T$). The denominators in Eq. (60) and (61) are given by hyperbolic functions at finite $T$. This gives a decay of $\exp(-2\pi T x/v_F)$ at large separations but interactions may give corrections to this. The $h^\pm$

functions behave as

$$h^{\pm}(\tau, x) = \pm \frac{T|x|}{4cv_F^2\sqrt{r}} + \text{finite}, \tag{102}$$

for large $|x|$. We now have

$$\mathbf{A} = -\frac{T|x|}{4cv_F^2\sqrt{r}} \begin{bmatrix} 1 & 1 \\ 1 & 1 \end{bmatrix} + \text{finite at large } x \tag{103}$$

$$\mathbf{B} = -\frac{T|x|}{4cv_F^2\sqrt{r}} \begin{bmatrix} 1 & -1 \\ -1 & 1 \end{bmatrix} + \text{finite at large } x. \tag{104}$$

These are once again negative semidefinite and proportional to $x$. This means that interactions can only contribute to a shorter decay length at finite $T$ except for the exceptional cases of interactions in the null spaces of $\mathbf{A}$ and $\mathbf{B}$ where it is unchanged. The long distance behavior of these correlators does thus always decay exponentially in the spatial separation. The static susceptibilities are then also finite since the $\tau$ integral is over a finite interval at finite $T$. No instabilities are present at finite $T$ and since correlators decay exponentially in the separation, they also do not show quasi-long-range order. Let us consider the static susceptibilities at finite $r$ in the $T \to 0$ limit. The $h^{\pm}$ functions behave as at $T = 0$ for separations $x \ll \min(v_F, c)/T, \tau \ll 1/T$ since the temperature only affects the boundary conditions of the theory (however, we show more rigorously that the asymptotic behavior is the same as at $T = 0$ up to $|x| \sim v_F/T$ in Appendix C). The correlators are exponentially suppressed for large spatial separations as $\exp(-2\pi x T/v_F)$ or faster and the time direction is periodic with periodicity $1/T$. This means that the finite temperature effectively introduces cut-offs in the Fourier transforms of the $T = 0$ static susceptibilities. We consider an explicit cut-off at $\sqrt{x^2/v_F^2 + \tau^2} < \Lambda \equiv \epsilon/T$ where $\epsilon$ is small such that the $T = 0$ correlators in Eq. (83) and Eq. (84) can be used up to this cut-off. The Fourier transform is largest at momenta where oscillations cancel, this happens at $q = 2k_F$ in the density susceptibility case and for $q = 0$ in the pair susceptibility case meaning that any potential divergences show up first at these momenta. Performing the cut-off Fourier transform at these momenta we find:

$$\chi_{\rho_\sigma, \Lambda}(q = 2k_F) \sim \begin{cases} \text{finite, for } \lambda_\sigma^- \lambda_\sigma^+ > -\pi v_F c \sqrt{r} \\ \log(\Lambda), \text{ for } \lambda_\sigma^- \lambda_\sigma^+ = -\pi v_F c \sqrt{r} \\ \Lambda^{-\frac{\lambda_\sigma^- \lambda_\sigma^+}{2\pi v_F c \sqrt{r}} - \frac{1}{2}}, \text{ for } \lambda_\sigma^- \lambda_\sigma^+ < -\pi v_F c \sqrt{r} \end{cases} \tag{105}$$

$$\chi_{b, \Lambda}(q = 0) \sim \begin{cases} \text{finite, for } \lambda_\uparrow^- \lambda_\downarrow^+ < 0 \\ \log(\Lambda), \text{ for } \lambda_\uparrow^- \lambda_\downarrow^+ = 0 \\ \Lambda^{\frac{\lambda_\uparrow^- \lambda_\downarrow^+}{2\pi v_F c \sqrt{r}}}, \text{ for } \lambda_\uparrow^- \lambda_\downarrow^+ > 0. \end{cases} \tag{106}$$

By finite we mean terms that are bounded as $T \to 0$. We expect the contribution from outside the cut-off to behave the same way since the integral is bounded in the $\tau$ direction and exponentially decays in the spatial directions. The full susceptibilities then behave like this:

$$\chi_{\rho_\sigma}(q = 2k_F) \sim \begin{cases} \text{finite, for } \lambda_\sigma^- \lambda_\sigma^+ > -\pi v_F c \sqrt{r} \\ \log(T), \text{ for } \lambda_\sigma^- \lambda_\sigma^+ = -\pi v_F c \sqrt{r} \\ T^{\frac{\lambda_\sigma^- \lambda_\sigma^+}{2\pi v_F c \sqrt{r}} + \frac{1}{2}}, \text{ for } \lambda_\sigma^- \lambda_\sigma^+ < -\pi v_F c \sqrt{r} \end{cases} \tag{107}$$

$$\chi_b(q = 0) \sim \begin{cases} \text{finite, for } \lambda_\uparrow^- \lambda_\downarrow^+ < 0 \\ \log(T), \text{ for } \lambda_\uparrow^- \lambda_\downarrow^+ = 0 \\ T^{-\frac{\lambda_\uparrow^- \lambda_\downarrow^+}{2\pi v_F c \sqrt{r}}}, \text{ for } \lambda_\uparrow^- \lambda_\downarrow^+ > 0, \end{cases} \tag{108}$$

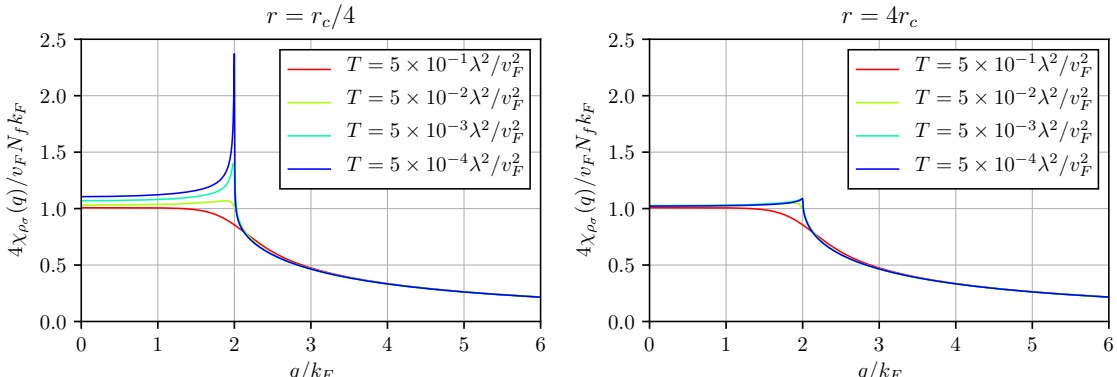

Figure 5: These figures show the static charge (or magnetic) susceptibility against momentum for $r$ in the region with an instability at $T = 0$ (left) and outside this region (right). The susceptibility has been calculated by a Fourier transform of (60) where the $h^{\pm}$ functions have been obtained numerically. Here we consider the somewhat artificial case of $\lambda_{\sigma}^{+} = -\lambda_{\sigma}^{-} \equiv \lambda$ in all directions such that we have a rotationally symmetric density correlator. We further use $k_F = 5\lambda^2/v_F^3$ and $c = v_F$. We see that the susceptibility diverges upon lowering temperature for $r < r_c$ and approaches a constant for $r > r_c$.

for low temperatures. The susceptibilities can easily be calculated numerically as well, see Figure 5 for the charge/spin susceptibility at different momenta.

# 4 Landau-damping corrections

The $N_f \to 0$ limit removes the corrections from the fermions onto the field $\phi$ such that it is still Gaussian and can be integrated out. However, we can add some corrections to the $\phi$ two-point function without making it non-Gaussian. These are important corrections to the IR at finite $N_f$ so being able to add these is interesting because it gives us an indication as to how $N_f$ corrections modify the results away from $N_f = 0$.

In addition to the $N_f \to 0$ limit have additionally considered the limit of large separations compared to $1/k_F$, which equivalently can be viewed as a large $k_F$ limit. The suppressed fermionic loops each come with a factor of $N_f k_F$ so these two limits do not commute and we cannot remove these loops if we take the limits in the opposite order. For full generality we may take these limits simultaneously keeping $N_f k_F$ constant and thus obtaining a new parameter. This was done in [32] and the limit was shown (with some caveats concerning the order of other limits) to suppress all symmetrized fermionic loops with more than two vertices, order by order in perturbation theory, and thus allowing us to still integrate out the order parameter fluctuations. A note should be made here: this limit gives a fermion with $k_x \sim \omega^{2/3}$ and when calculating non-planar diagrams that appear at order $\lambda^4$ and higher, it is then necessary to keep the Fermi surface curvature term $-\beta k_y^2$ in the fermion propagator when the external energy is small $\omega \lesssim \beta^3 N_f^2 k_F^2 \lambda^4/v_F^4 c^4 \equiv \omega_\beta$ [42]. For our spherical Fermi surface we have $\beta = v_F/(2k_F)$ and thus

$$\omega_\beta \sim \frac{N_f^2 \lambda^4}{c^4 v_F k_F}. \tag{109}$$

This energy scale vanishes in the double limit we consider, so while our results are valid in this

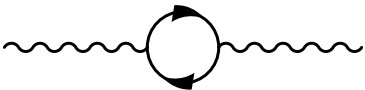

Figure 6: A two-vertex fermion loop giving corrections to the field $\phi$.

limit, one should note that the full theory contains more fermion loops, but also curvature corrections to the non-planar diagrams below this energy scale. Note that this scale is also much smaller than the scale set by the coupling, $\omega_B \ll \lambda^2/c^2$ when $\lambda^2/c^2 \ll v_F k_F$, also for $N_f = 1$. The double limit thus removes corrections from fermion loops with more than 2 vertices, and suppresses the energy scale where non-planar diagrams receive curvature corrections.

Fermion loops are also suppressed in the matrix large $N$ limit studied in several works mentioned in the introduction. A similar double limit reintroducing some of the effects from fermion loops was studied in [21].

We would like to consider the same $N_f \to 0$, $k_F \to \infty$, $N_f k_F = $ constant limit as in [32] but for the systems and correlation functions we are studying here. In this work we allow the coupling function to depend on momentum and we consider finite temperatures so we need to verify that the same suppression of fermionic loops with more than two vertices remains. This can actually already be seen concerning corrections to the $\phi$ two-point function from our above calculated density correlator. The field $\phi$ couples to the fermion densities so the amputated $\langle \phi \phi \rangle$ correlator is given by $\langle \rho \rho \rangle$. From Eq. (60) we see that $\langle \rho \rho \rangle$ is composed of a term with small momenta (compared to $k_F$) and a term with momenta of order $k_F$. Only the former term is relevant since the low-energy $\phi$ modes have $Q = 0$. This term is independent of the couplings and is thus simply what is obtained from the 2-vertex fermion loop. To verify suppression of higher $n$-point $\phi$ corrections in this limit we explicitly calculate such fermion loops in Appendix D.

Having verified this cancellation we may consider what this limit entails for the correlation functions we are interested in. The earlier strict $N_f \to 0$ limit lets us use the free boson correlation function when evaluating the $h^\pm$-functions. Now we should calculate the $h^\pm$ functions with a $\phi$ two-point function that is corrected be 2-vertex fermion loops, see Fig. 6. The $N_f \to 0$ limit was also used to suppress diagrams of the type in Fig. 1(b) contributing to the density-density correlator. Those diagrams come with an extra factor of $k_F$ as opposed to Fig. 1(a) and can seemingly not be neglected in the double limit we now consider[3]. The diagrams of the type in Fig. 1(b) can not simply be found by using the background field Green's function since we can only calculate it for insertions separated a distance $x \gg 1/k_F$ and here we would need to have two insertions at the same point. We thus need to evaluate these diagrams some other way. Luckily, this class of diagrams turns out to be manageable. Let us first consider such diagrams in momentum space, with a large momentum incoming to the density vertex $q \gtrsim k_F$. The momenta going through the bosons connecting the two fermion loops are now of order[4] $k_F$. The propagators are of the form $1/(\omega^2 + c^2 q^2 + r^2 - \Pi)$ so these diagrams get a further $k_F$ suppression and need not be considered.

Next consider the case of a small momentum incoming to the density vertex, $q \ll k_F$. In the above we found that all loops with more than 2 vertices cancel to leading order in large $k_F$. This assumed all incoming momenta were small compared to $k_F$ and it is now applicable.

---

[3]I thank Chris Hooley and Andriy Nevidomskyy for a discussion where this came up.

[4]This is true at any finite order in perturbation theory. At large perturbative order $n$ the large momentum is spread among many boson exchanges and non-perturbative effects may allow such processes as is discussed in Chapter 1.3.8 of [37]. Now we nevertheless work in the less attractive limit as in [32] where we consider $k_F$ large before we sum all diagrams and this is not an issue.

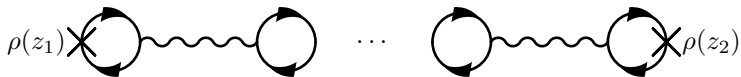

Figure 7: This figure shows a class of bubble chain diagrams that contribute in the combined limit $N_f \to 0$, $k_F \to \infty$ that is not captured by the above framework and has to be calculated separately. The ellipsis indicate that this class of diagrams has any number of bubbles, larger than 1.

We thus have that the only diagrams contributing to leading order are the ones with a single boson connecting the left and right fermion loop in Fig. 1(b) such that both loops each have two vertices. The boson being exchanged however receives corrections from further 2-vertex fermion loops so when $q \ll k_F$ we must sum all diagrams of the form shown in Fig. 7. The first and last vertices are momentum independent for the density-density correlator. All other vertices come with a momentum dependent coupling constant $\lambda(\theta)$. We write the two-vertex fermion loop as $\Pi_2[f](\omega, \mathbf{q})$ where $f_\sigma(\theta)$ is the combined momentum and spin dependence of the two vertices. We have that the summed contributions from diagrams of the type in Fig. 7 is given by

$$\langle \rho\rho(q) \rangle_{\text{bubbles}} = \Pi_2[1](\omega, \mathbf{q}) + \Pi_2[\lambda(\theta)](\omega, \mathbf{q})^2 \sum_{n=1}^{\infty} \frac{\left(\Pi_2[\lambda(\theta)^2](\omega, \mathbf{q})\right)^{n-1}}{(\omega^2 + c^2\mathbf{q}^2 + r^2)^n}. \tag{110}$$

We now need to calculate the two-vertex fermion loop $\Pi_2[f](\omega, \mathbf{q})$ both for this additional contribution to the density correlation function and to be able to calculate the $h^\pm$ functions.

$\Pi_2[f](\omega, \mathbf{q})$ was calculated at a finite temperature in $d = 3$ in [33]. They found that the result is identical to the zero temperature result and this is true also in $d = 2$. Only fermions close to the Fermi surface contribute for $\phi$ momenta small compared to $k_F$ and we can approximate (see Appendix E) this loop by the integral

$$\tilde{\Pi}_2[f](\omega_n, \mathbf{q}) = -\sum_{i,\sigma}^{N_f, \{\uparrow, \downarrow\}} \frac{ik_F \omega_n}{(2\pi)^2 v_F} \int \mathrm{d}\theta \frac{f_\sigma(\theta)}{i\omega_n - v_F q \cos(\theta)}, \tag{111}$$

where $\theta$ is an angle measured from the external momentum $\mathbf{q}$ and $f_\sigma(\theta)$ is the momentum dependence of the vertices evaluated at the Fermi surface: $f_\sigma(\theta) = f_\sigma(k_F \hat{n}(\theta))$. To solve this we expand the vertex momentum dependence in Fourier series

$$f_\sigma(\theta) = \sum_{n=0}^{\infty} f_{\sigma,n} \cos(n\theta) + \sum_{n=1}^{\infty} \tilde{f}_{\sigma,n} \sin(n\theta). \tag{112}$$

Calculating the sums and integrals (see Appendix E) we have

$$\tilde{\Pi}_2[f](\omega_n, \mathbf{q}) = -2N_f \frac{k_F |\omega_n|}{2\pi v_F \sqrt{q^2 v_F^2 + \omega_n^2}} \sum_{\sigma,m=0}^{\infty} f_{\sigma,m} \left( \frac{-i\,\mathrm{sgn}(\omega_n) q v_F}{\sqrt{q^2 v_F^2 + \omega_n^2} + |\omega_n|} \right)^m. \tag{113}$$

Now we consider the boson self-energy and replace $f_\sigma(\theta) = \lambda_\sigma^2(\theta)$. The self-energy dominates over the $\omega^2$ in the (critical) bosonic propagator and the low-energy scaling is

$$\omega N_f k_F \lambda^2 \sim c^2 v_F^2 q^3. \tag{114}$$

The fraction in the power can then be replaced by $-i\,\mathrm{sgn}(\omega_n)$ in this limit. We then find

$$\tilde{\Pi}_2(|\omega_n| \ll qv_F) = -2N_f \frac{k_F|\omega_n|}{2\pi v_F^2 q} \sum_{\sigma,m=0}^{\infty} \lambda_{\sigma,m}^{(2)}(-i\,\mathrm{sgn}(\omega_n))^m \tag{115}$$

$$= N_f \frac{k_F}{\pi v_F^2 q} \sum_{\sigma} \left( -|\omega_n| \frac{\lambda_\sigma(\frac{\pi}{2})^2 + \lambda_\sigma(-\frac{\pi}{2})^2}{2} + i\omega_n\,\mathrm{PV}\int_0^{2\pi} d\theta \frac{\lambda_\sigma(\theta)^2}{2\pi\cos(\theta)} \right). \tag{116}$$

Assuming $\lambda_\sigma(\theta)$ is parity symmetric or antisymmetric we find exactly the result of [15]: to leading order the boson only couples to fermions with perpendicular momenta. Interestingly, for a non-centrosymmetric theory where $\lambda_\sigma(\mathbf{k})^2 \neq \lambda_\sigma(-\mathbf{k})^2$, we find that this is not the case and there is an extra contribution from the boson coupling to fermions all around the Fermi surface given by the principal value integral in Eq. (116). This extra contribution is odd in both frequency and momentum (the angle $\theta$ was defined w.r.t. the incoming momentum $\mathbf{q}$). Analytically continuing this to Minkowski signature we find that it is not a damping term but instead it contributes a singular real part to the polarization. We are not aware if this extra term appearing in non-centrosymmetric systems has been studied before but it is outside the scope of this paper so it is left for future studies. We only consider centrosymmetric systems henceforth.

Now that we have calculated the boson self-energy we first use this to calculate the $\langle\rho\rho(q)\rangle_{\mathrm{bubbles}}$ contribution (see Eq. (110)) to the density-density correlation function. Eq. (110) is written in momentum space but we are interested in correlators in real space at large separations and thus Fourier transform this. We stay away from criticality and consider distances longer than $c/\sqrt{r}$. We do this in Appendix F (for a constant $\lambda(\theta) = \lambda$) and find:

$$\langle\rho\rho(\tau,x)\rangle_{\mathrm{bubbles}} = \mathcal{O}\left( \frac{1}{(x^2 + v_F^2\tau^2)^{3/2}} \right), \tag{117}$$

for large $x$ and $\tau$ and this contribution does thus not affect any conclusions regarding instabilities so we need not consider this more.

Now we use a corrected $D$ when calculating $h_{\mathrm{LD}}^\pm$:

$$D_{\mathrm{LD}}(\omega_n, \mathbf{q}) = \frac{1}{\omega_n^2 + c^2 q^2 + r - \Pi_2(\omega_n, k)}. \tag{118}$$

We use the subscript LD to indicate that $D$ and $h^\pm$ have been calculated with the above Landau-damping correction. This calculation is performed in Appendix G. We calculate the long distance behavior of $h_{\mathrm{LD}}^\pm(\tau = 0, x)$ to see if the instabilities found in the previous section have changed. Away from criticality ($r > 0$), at both $T = 0$ and $T > 0$ we find that only the finite part of $h_{\mathrm{LD}}^\pm(\tau = 0, x)$ has changed compared to $h^\pm(\tau = 0, x)$. The growth at long distances is the same and the conclusions about instabilities are thus unchanged by the inclusion of resummed 1-loop Landau-damping corrections. However, right at the QCP where $r = T = 0$ we find that the $h_{\mathrm{LD}}^\pm(\tau = 0, x)$ functions no longer grow linearly in the separation but now grow with a power-law:

$$h_{\mathrm{LD}}^+(0, x) = -|x|^{1/3} \frac{\Gamma\left(\frac{2}{3}\right)}{3\sqrt{3}\pi M_D^{2/3} v_F^{4/3}} + \mathrm{finite} \tag{119}$$

$$h_{\mathrm{LD}}^-(0, x) = -3h_{\mathrm{LD}}^+(0, x) + \mathrm{finite}. \tag{120}$$

Note that there are now two scales at the QCP: $\lambda^2$ and $M_D$ so the $h_{\mathrm{LD}}^\pm(\tau = 0, x)$ functions only have this behavior asymptotically. Since the $h_{\mathrm{LD}}^\pm$ functions both have the same power-law

growth we have to consider them both when considering instabilities. Analogously to the non-Landau damped case we find that unbounded growth of correlators is governed by Eq. (77) and $R$ is now independent of all parameters and given by $R = 1/\sqrt{2}$.

# 5 Results for specific systems

In the following subsections we use the above results to study specific fermionic systems near QCPs to obtain their $N_f \to 0$ phase diagrams. We start each subsection with a brief description of the quantum phase transition and how its fluctuations couple to the fermions, i.e. what the structure of $\lambda_\sigma(\theta)$ is. We finally also consider another system that can be described by the same model, fermions coupled to an emergent gauge field. While the study of these models is motivated by experimental systems, it is not very useful to make a direct comparison with our results since we consider very simplified models of them. Instead we focus on some earlier theoretical results on the same and similar models when making comparisons.

## 5.1 Charge nematic transition

In this subsection, we consider an $l = 2$ charge nematic quantum critical point such as the ones that have been observed in the Cu- [1,2] and Fe- [3–5] based SCs. These QCPs are due to electronic instabilities towards a state where the Fermi surface shape breaks the $C_4$ rotational symmetry of the lattice. The resulting nematic state has a remaining $C_2$ rotational symmetry and symmetry under spin inversion. The order parameter is a single real scalar field that is odd under $\pi/2$ rotations. This means that $\lambda_\sigma(\mathbf{k})$ is independent of $\sigma$ and has $d$-wave symmetry. In the analysis of the previous sections we considered correlators of fermion bilinears separated spatially in a chosen direction $\hat{n}$. For a particular $\hat{n}$ we then have $\lambda_\uparrow^+ = \lambda_\downarrow^+ = \lambda_\uparrow^- = \lambda_\downarrow^- \equiv \lambda_{\hat{n}}$ where $\lambda_{\hat{n}}$ has $d$-wave symmetry and thus is 0 in four different directions, the so-called cold spots. For $T = 0$ and $r > 0$, we find from Eq. (83) that nematic fluctuations suppress the charge and spin correlators at long distance:

$$\langle \rho_\alpha(0,0)\rho_\alpha(0,x\hat{n})\rangle = C_1 N_f k_F \sin(2k_F x)\left(\frac{1}{x}\right)^{3+\frac{\lambda_{\hat{n}}^2}{2\pi c v_F \sqrt{r}}} - \frac{N_f k_F}{4\pi^3 x^3} + \text{subleading}. \tag{121}$$

This is identical to what happens near a ferromagnetic QCP so we defer the discussion of this to the next subsection. From Eq. (108) we find that the nematic fluctuations lead to enhanced pairing fluctuations as $T = 0$ is approached, most strongly near the QCP:

$$\chi_b(q=0) \sim T^{-\frac{\lambda_{\hat{n}}^2}{2\pi v_F c \sqrt{r}}}. \tag{122}$$

Instead of a logarithmic growth of fluctuations, now fluctuations grow with a power-law as $T \to 0$ and furthermore, the exponent diverges as the QCP is approached. As noted in the previous section, these results are unchanged by the inclusion of Landau-damping corrections. Note that we have kept the boson gap fixed here, other works that do not simply postulate the quantum critical action, but derive it from a model with a Pomeranchuk instability, find a temperature dependent gap [40,43] and thus a more complex temperature dependence here.

While we have considered $s$-wave pairs, this result does not necessarily mean that the ground state is an $s$-wave SC. We could similarly consider e.g. $p$ and $d$-wave pairs that contain spatial derivatives and different spin structures. This can be investigated within the framework we have used here but due to spatial derivatives breaking rotational symmetry it will be a longer calculation that we leave for the future. Instead, we make an educated guess of the

result. The spin is unimportant in this case since $\lambda_\uparrow(\mathbf{k}) = \lambda_\downarrow(\mathbf{k})$. The conclusions regarding instabilities come from the asymptotic behavior of the correlation functions. We have derivatives acting on the background field Green's function of Eq. (38) for non-$s$-wave pair correlators. The derivative may hit any of the factors in Eq. (34). Hitting the exponent $-isk_F|x_{12}|$ only brings down a constant and the resulting term does thus have the same asymptotics as for the $s$-wave case. While it is possible some cancellation occurs for e.g. $p$-wave, this is not expected to be universal and happen for states of all symmetries apart from $s$-wave. This indicates that whether the pairing instability we found here results in $s$-wave or a gap of some other symmetry is something we cannot tell using this analysis. It seems likely several of these correlators diverge at the same points in the phase diagram and the groundstate symmetry is determined by the next to leading order $N_f$ and $1/k_F$ corrections and higher order terms in the action. Since the coupling $\lambda_{\hat{n}}$ has a $d$-wave form factor and thus is 0 on four nodes at the Fermi surface, there will be directions in which the instabilities are not seen. This allows for nodes or lukewarm regions along the Fermi surface where the gap is 0 or small and thus compatible with a $d$-wave state.

We have found that nematic fluctuations lead to enhanced superconductivity in the $N_f \to 0$ limit but the symmetry of the gap is not definite. This is consistent with studies of the charge nematic transition in two dimensions in the matrix large-$N$ limit [23] where nematic fluctuations enhance SC but also there the symmetry of the SC groundstate is non-universal. The matrix large-$N$ limit has also been employed on the charge nematic transition in $3 - \epsilon$ dimensions [22, 24] where $\epsilon$ is small. There the order of the $\epsilon \to 0$ and $N \to \infty$ limits decide whether the system is a SC or not [24]. [44] studies the same model of a nematic QCP (at $N_f = 1$) as us and additionally include four-fermion interactions. By considering small interactions and staying a small distance away from the QCP they can integrate out the nematic fluctuations and get a correction to the four-fermion interaction, essentially staying within a Fermi liquid framework. They solve the gap equation and find that this correction gives an increase in $T_c$ compared to the BCS case as criticality is approached. Like us, they find that nematic fluctuations increase susceptibility towards pairing of both $s$-wave and $d$-wave symmetry and find that the bare four-fermion interaction is what decides the symmetry of the resulting superconducting state and not the near-critical nematic fluctuations.

The authors of [45] include one-loop self-energy corrections to both the boson and the fermions and argue that this allows them to access the strongly coupled regime at the QCP. The order parameter field is static in their case but they are otherwise in the same limit as we have assumed; they consider $l = \lambda^2/v_F k_F$ a small parameter. They find that nematic fluctuations lead to superconductivity, with a finite $T_c$, and they also find that the SC symmetry is non-universal.

Finally we mention that the two-dimensional charge nematic transition has been studied using sign-problem-free determinant quantum Monte-Carlo (DQMC). This was done by putting fermions on a finite lattice and coupling them to pseudospin-1/2 degrees of freedom [46,47]. The spins have a nematic QCP whose fluctuations give strong corrections to the fermions. The first work does not find any pairing instability however they find large pairing fluctuations. In the second study the authors increase the coupling between the spins and the fermions beyond what is physical in a microscopic interpretation of the model and they instead regard it as an effective model. They then find a superconducting dome covering the QCP with $s$-wave symmetry in the symmetric phase.

As opposed to these earlier studies, we only find a pairing instability at $T = 0$. However, we believe this is expected due to the Mermin-Wagner theorem. By departing from the strictly two-dimensional case and including four-fermion interactions at finite $N_f$ we expect an increase in $T_c$ compared to that of BCS theory since we find an increase in pairing susceptibility due to nematic fluctuations. This is something we leave for future works to study.

## 5.2 Ferromagnetic and spin nematic transition

Ferromagnetic and spin nematic ordering both break spin-inversion symmetry. Electrons spontaneously order their spins in the same direction in a ferromagnetic phase.

The spin-nematic phase is similar to the charge-nematic case, the Fermi surface spontaneously breaks rotational symmetry, but in perpendicular directions for spin up and down. In fact, all of the orders considered here can be seen as coming from Pomeranchuk instabilities. Ferromagnetic ordering is simply the Fermi surface of one spin spontaneously becoming larger than for the opposite spin.

There will be soft fluctuations in the order parameter in the case where these transitions are continuous. The fluctuations couple with opposite sign to fermions of opposite spin, $\lambda_\uparrow(\mathbf{k}) = -\lambda_\downarrow(\mathbf{k})$, for both the ferromagnetic and the spin-nematic transition. However, $\lambda_\uparrow(\mathbf{k})$ has $s$-wave symmetry in the former case and $d$-wave symmetry in the latter. For a chosen spatial direction $\hat{n}$ we then have $\lambda_\uparrow^+ = \lambda_\uparrow^- = -\lambda_\downarrow^+ = -\lambda_\downarrow^- \equiv \lambda_{\hat{n}}$ for the coupling of both ferromagnetic and spin-nematic fluctuations. Their effects on the fermions will thus be the same when studied using the framework of this paper, with the caveat that in the spin-nematic case there are four coldspots in $\lambda_{\hat{n}}$ and none in the ferromagnetic case.

Plugging in the coupling function in Eq. (77), we do not find any instabilities at the QCP. In fact, we find that components of the leading small $N_f$ CDW/SDW fluctuations and pairing fluctuations receive faster power-law fall-offs at $T = 0$ due to interactions:

$$\langle \rho_\alpha(0,0)\rho_\alpha(0,x\hat{n})\rangle = C_1 N_f k_F \sin(2k_F x)\left(\frac{1}{x}\right)^{3+\frac{\lambda_{\hat{n}}^2}{2\pi c v_F \sqrt{r}}} - \frac{N_f k_F}{4\pi^3 x^3} + \text{subleading} \tag{123}$$

$$\langle b^\dagger(0)b(0,x\hat{n})\rangle = 2N_f k_F C_2 \left(\frac{1}{x}\right)^{3+\frac{\lambda_{\hat{n}}^2}{2\pi c v_F \sqrt{r}}} + 2N_f k_F \,\mathrm{Im}\left(C_4 \frac{e^{-i2k_F x}}{x^3}\right) + \text{subleading}. \tag{124}$$

This means that the pair susceptibility is now finite at $T = 0$ (see Eq. (108)) and the divergence in the free theory is cured by interactions with the ferromagnetic or spin-nematic fluctuations. We thus have a naked QCP with strong NFL behavior but suppressed pairing and suppressed Friedel oscillations.

At finite temperatures we find that the correlations lengths $l_{\hat{n}}$ of the above components receiving corrected power-laws also decrease due to interactions (see Eq. 60–61):

$$l_{\hat{n}}^{-1} = \frac{T}{v_F}\left(2\pi + \frac{\lambda_{\hat{n}}^2}{c v_F \sqrt{r}}\right). \tag{125}$$

We have been considering the leading contribution to the correlators at $x \gg 1/k_F$. With these corrections due to interactions we find a faster decay in separation $x$ for some of the terms above so it is possible that the subleading behaviors that we have not considered are actually dominating these. To be perfectly consistent we should then remove these terms above and regard them as part of the neglected terms that are subleading at long distances. The leading long-range behavior does thus not necesarily behave as described by Eq. (123)–(125) at large separations. The above asymptotic result should be taken with caution and conservatively viewed as simply a cancellation of the diagrams at leading order for large $x$.

Fermions in 2D coupled to ferromagnetic fluctuations were studied in [48] using DQMC. The authors consider a system with two flavors (orbitals) of spinful fermions and find spin triplet SC tendencies. By adding an orbital index to the fermions we may also consider orbit-singlet spin-triplet pairing instead of the spin-singlet pairing studied so far in this work. We define the spin-triplet pair operators

$$b_{t,\sigma} = \psi_{1,\sigma}(x)\psi_{2,\sigma}(x). \tag{126}$$

We can calculate the pair correlator the same way as in the case of the spin-singlet pair, the only difference is that we never encounter $\lambda_\sigma(\mathbf{k})$ with mixed spins, the result is the same as what is found for the spin-singlet pair correlator near the nematic QCP studied in the previous subsection. This means that in a two-orbital system coupled to ferromagnetic fluctuations, we also find a pairing instability in the spin-triplet channel in the $N_f \to 0$ limit.

## 5.3 Circulating current transition

The circulating current phase proposed by Simon and Varma [49] breaks lattice $C_4$ rotational symmetry and time-reversal symmetry. It requires a two component order parameter $\phi_a = (\phi_x, \phi_y)$ that couples to the fermions through a coupling function with $p$-wave symmetry independently of spin: $\lambda_{a,\sigma}(\mathbf{k}) \propto \mathbf{k}_a$ [15]. As we only consider two antipodal patches in the direction $\hat{n}$ at a time we may simply consider a single $\phi$ corresponding to $\phi_a$ projected in the $\hat{n}$ direction. We then have $\lambda_\uparrow^+ = \lambda_\downarrow^+ = -\lambda_\uparrow^- = -\lambda_\downarrow^- \equiv \lambda_{\hat{n}}$ and no cold-spots despite the $p$-wave symmetry since we actually have a two-component order parameter. Again considering Eq. (77), we find that fluctuations at the circulating current QCP lead to an instability. This time in the charge/spin density channel at momentum $Q = 2k_f$. Departing from the QCP but still at $T = 0$, we find that the divergence in the charge and spin susceptibilities persists up to a critical order parameter fluctuation gap $r_c$, see Eq. (89):

$$r_c = \frac{\lambda_{\hat{n}}^4}{\pi^2 c^2 v_F^2}. \tag{127}$$

Density fluctuations diverge upon approaching the $T = 0$, $0 < r < r_c$ interval either from finite temperatures or larger gaps. A phase diagram is shown in Fig. 8. The interval terminates at a new QCP where there now are large fluctuations at wave-vector $Q = 2k_F$. These fluctuations may give rise to important finite $N_f$ corrections, something we leave for the future to consider.

As in the above ferromagnetic case, we find that $s$-wave pairing is suppressed and there is no superconducting instability at $T = 0$. In the ferromagnetic case this was due to the coupling acting with different signs on the electrons with opposite spins making up a Cooper pair and we saw that it is still possible to have triplet pairing in the ferrromagnetic case. In this case the coupling is independent of spin, but it couples with opposite sign to fermions on opposite sides of the Fermi surface and the coupling does not induce triplet pairing in this case.

## 5.4 Spin liquids: Emergent gauge field

As mentioned in the introduction, Eq. (1) can also describe fermions coupled to a $U(1)$ gauge field $A$ which appears in models of spin liquids. The electric potential $A_0$ is screened and can be neglected [14]. By defining $\phi_a = \mathbf{A}_a$ we can capture the interaction of the magnetic potential and the fermions through our action Eq. (1) with the following choice of coupling function

$$\lambda_a(\mathbf{k}) = e\frac{\mathbf{k}_a}{m}, \tag{128}$$

where $e$ is the coupling strength. The longitudinal part of $\mathbf{A}_a$ is set to 0 in the Coulomb gauge ($\mathbf{k}\cdot\mathbf{A} = 0$). Once we pick a direction $\hat{n}$ and only consider patches $k = \pm k_F \hat{n}$ we may thus neglect $A_{\hat{n}}$ and simply call the transverse component $\phi$. We thus find that we have a similar coupling function as in the case of a circulating current phase transition: $\lambda_\uparrow^+ = \lambda_\downarrow^+ = -\lambda_\uparrow^- = -\lambda_\downarrow^- = ev_F$. Also in this case, there are no cold-spots since we picked the direction $\hat{n}$ arbitrarily. At $T = 0$ we thus find that this system also has an instability towards forming charge/spin density waves around $Q = 2k_F$. Now $r = 0$ since $\phi$ is massless by gauge invariance and this instability is present for arbitrarily weak but non-zero interaction strength $e$. This result is opposed to what

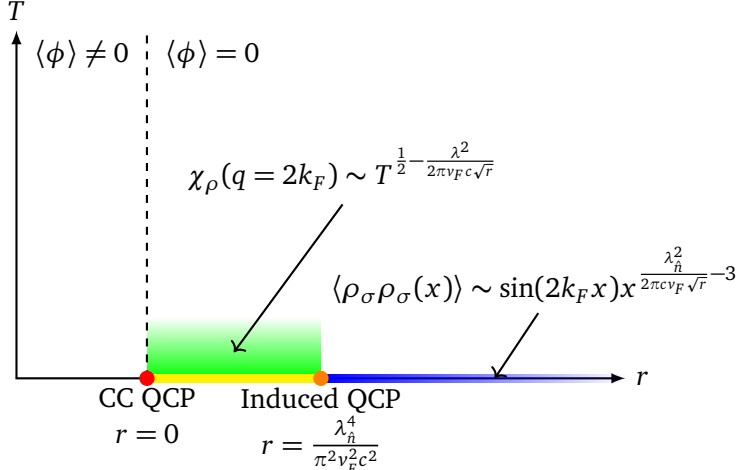

Figure 8: Sketch of the $N_f \to 0$ phase diagram in the vicinity of a circulating current (CC) QCP (red dot). The ordered phase (negative $r$) is not considered. Order parameter fluctuations near the CC QCP lead to an instability towards forming either charge- or spin-density waves at $T = 0$. This happens for an interval (yellow) of fluctuation gaps terminating at $r = r_c$ where there is an induced QCP (orange dot). Approaching this interval from above leads to a diverging susceptibility indicated in the green region.

is found in the (vector) large $N_f$ limit [14] where no antiferromagnetic instability is found at $T = 0$. This model at $T = 0$ has also been studied at matrix large $N$ in [18]. The authors find that an instability at $2k_F$ in the charge density is present at large $N$, but suspect that the point of interest $N = 2$ may be stable. They proceed by considering a modified boson dynamical critical exponent $z_b = 2 + \epsilon$ (in our model $z_b = 3$) and a combined limit where both $N \to \infty$ and $\epsilon \to 0$. $\epsilon N$ tunes the relative strength of the quasiparticle smearing that suppresses instabilities and what they call "Amperean attraction" which enhances it. Taking these limits, they find that whether the theory is unstable depends on the order of the two limits. Whether the physically interesting case of $N = 2$, $\epsilon = 1$ is unstable then depends on what the $\epsilon$–$N$ phase diagram looks like.

Now we consider finite temperatures. The apparent IR divergence discussed in the paragraph after Eq. (101) becomes relevant for this system since $r = 0$. We find that the $N_f \to 0$ spin liquid has the leading large distance pair correlator completely suppressed at finite temperatures because of the divergence in the $h^\pm$ functions and it is entirely composed of the subleading terms we have not considered in this work. Since the divergence is due to the $n = 0$ Matsubara sum term, it is unaffected by the inclusion of Landau-damping.

The $n = 0$ terms in the $h^\pm$ functions cancel out in the charge (and spin) density correlator exponent and it decays exponentially at long distances for all finite temperatures. Its precise form is found by calculating the finite piece of the $h^\pm$ functions. However, since the correlator grows exponentially in separation at $T = 0$ (or as $\langle \rho\rho \rangle \sim \exp(Ax^{1/3})$ when including Landau-damping, see Eq. (119), (120)), we expect the charge and spin susceptibilities to diverge faster than a power-law as $T = 0$ is approached and the $\tau, x$ cut-offs move to infinity as $v_F/T$, see end of Section 3.4.

## 5.5 Multiple critical points

We can easily consider a system close to multiple QCPs of different natures. Several types of order parameter fluctuations give corrections to the fermions that have to be treated simulta-

neously. As we see in Eq. (60) and Eq. (61), these corrections simply sum in the exponents. We have seen interaction effects both suppress and enhance the long distance correlation functions so it is possible that the instability induced at $T = 0$ near one QCP is cured by the vicinity to another. Whether an instability shows up or not depends on the relative strengths of the interactions in the long distance limit. This depends on the coupling strengths that are direction dependent so we consider a direction $\hat{n}$ at a time. We find that the relevant strength of a QCP is of the form

$$\alpha_a = \frac{\lambda_{a,\hat{n}}^2}{c_a v_F \sqrt{r_a}}, \tag{129}$$

where $a$ refers to the index on the corresponding order parameter field $\phi_a$ and $\lambda_{a,\hat{n}}$ are the relevant components of the interaction strength as defined in the above subsections. These strengths are unaffected by the inclusion of Landau-damping corrections away from the QCP. We find the following conditions for CDW/SDW and pairing instabilities:

$$\alpha_{CC} - \alpha_{CN} - \alpha_{FM} - \alpha_{SN} > \pi \implies \text{CDW/SDW instability} \tag{130}$$

$$\alpha_{CN} - \alpha_{CC} - \alpha_{FM} - \alpha_{SN} > 0 \implies \text{Spin-singlet pairing instability}, \tag{131}$$

where CN, FM, SN, CC refers to charge nematic, ferromagnetic, spin nematic and circulating current transitions, respectively. The effects of the different fluctuations are as expected from the above subsections. $\alpha_a^{-1}$ can be viewed as the effective distance to the QCP labeled $a$. A three-dimensional phase diagram is shown in Fig. 9 with distances to charge nematic, ferromagnet and circulating current transitions on the axes.

Noting that the strengths $\alpha_a$, as defined, are always non-negative, we find that there is no phase competition between the pairing state and the CDW/SDW states, only one of the conditions above are satisfied at a time. However, we have only considered a single direction $\hat{n}$. The nematic fluctuations have cold spots and this means that we may have systems with phase competition. I.e. they may show a CDW/SDW type instability when considering one part of the Fermi surface and a pairing instability when considering another part and we cannot resolve within the current framework which one is winning.

Phase non-coexistence/competition of antiferromagnetism (AF) and SC is also seen in unconventional SCs with AF $T_c$ going to 0 at the point where SC $T_c$ is optimal [5, 50, 51]. While this may be indicative of AF fluctuations near an AF QCP enhancing superconductivity, we find that another possible mechanism giving a similar result may be seen for an AF phase in the vicinity of a nematic QCP. Pairing is most strongly enhanced right at the nematic QCP and at the same point we find that the exponent in the power-law decay of the normal-state AF correlator goes to $-\infty$. It is not entirely clear how the interactions giving rise to the AF phase would interact with the nematic fluctuations. However, if those interactions can be written in the form of gapped CC fluctuations, then we find from the above that they are completely outdone by the fluctuations at a nematic QCP and the AF state should disappear at or before the nematic QCP where $\alpha_{CN} = \infty$.

# 6 Discussion

In this paper we have investigated instabilities of fermions in two dimensions coupled to order parameter fluctuations near different QCPs and emergent gauge fields, in the $N_f \to 0$ limit. Taking this limit allowed us to calculate charge, spin and pair correlation functions at long distances, $x \gg 1/k_F$. First we considered a general model without specializing to a particular system. While we were agnostic to the momentum dependence of the coupling functions, we assumed an otherwise rotationally symmetric Fermi surface for convenience.

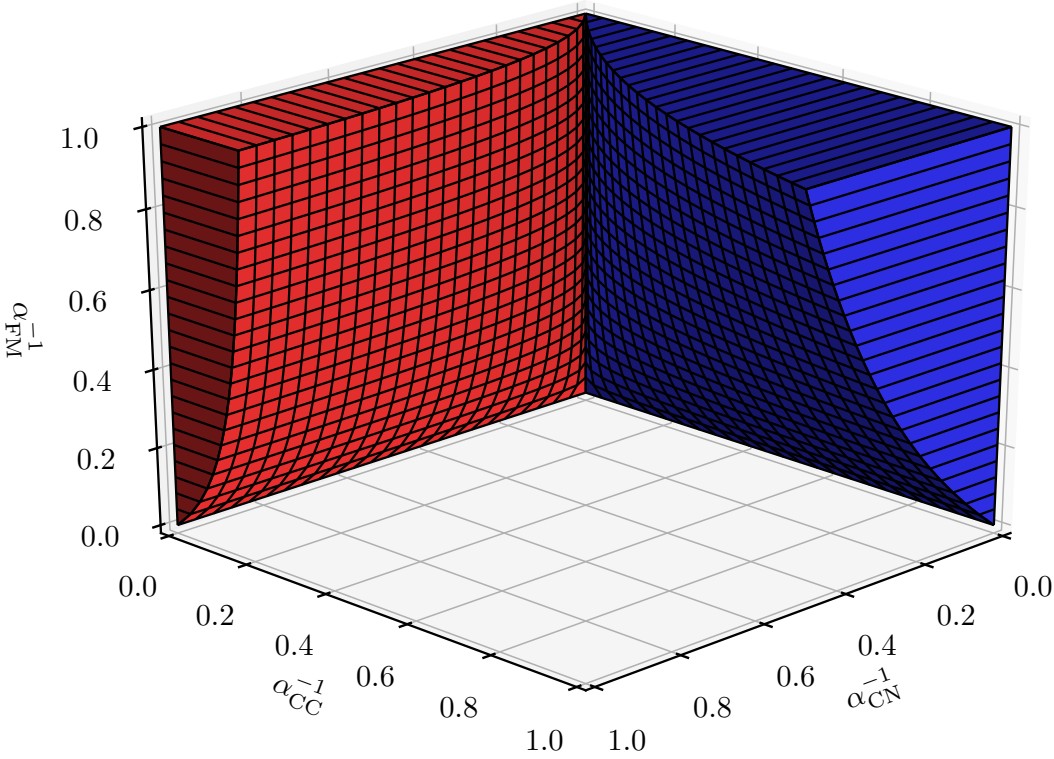

Figure 9: This shows a $T = 0$ phase diagram for $N_f = 0$ fermions coupled to three different types of order parameter fluctuations. The three axes show the relative strength of the order parameter fluctuations as defined by Eq. (129). The red region shows an instability towards CDW/SDW and the blue region shows an instability towards pairing. The empty region shows neither of these instabilities in the density or pairing correlation functions. The vicinity of the CC and FM QCPs inhibit pairing while the vicinity of the CN and FM QCPs inhibit the CDW/SDW instability.

We found expressions for correlators in terms of functions we label $h^{\pm}$. They are responsible for interactions between fermions within a patch, and for interactions of fermions in opposite patches, respectively. This means that $h^{+}$ is solely responsible for quasi-particle smearing effects while the attractive glue between fermions giving rise to pairing, CDW or SDW is generated by $h^{-}$. We find an inequality, Eq. (67), that settles that the former effect is stronger.

We considered both gapped and critical fluctuations and zero and finite temperatures. We found that an IR divergence in perturbation theory at finite temperatures for critical fluctuations was cured by summing all diagrams in the small $N_f$ limit.

All considered correlation functions show exponential decays at large separations at finite $T$ and power-law behavior at $T = 0$, for finite gaps. We found that there are cases where the powers are negative such that correlation functions show unphysical asymptotic growths at long distances, but also milder cases with power-law decay where nonetheless static susceptibilities diverge. We interpreted this as instabilities towards states where the corresponding operator condenses at $T = 0$. We proceeded by calculating the behavior of susceptibilities as $T = 0$ is approached from finite $T$.

Next we considered a modified boson propagator that includes the one-loop Landau-damping correction. We show that also in this more general setup, as compared to

[32], we can interpret this as the same controlled approximation in the double limit $N_f \to 0, N_f k_F = $ const. We find the inclusion of fermion loops with two vertices does not change the asymptotic correlator decay lengths at finite $T$ nor the exponent in the power-law at $T = 0$. However, the long distance behavior right at the QCP was shown to be affected by the inclusion of Landau-damping corrections and the constant prefactor and subleading terms are affected by Landau damping.

The diverging susceptibilities are all proportional to $N_f$ which is taken to 0 here so one may question whether the divergences indicate instabilities since we find them in the limit $N_f \to 0$. We argue that for the purpose of studying the $N_f \to 0$ limit to get insights about the finite $N_f$ case, these divergences should be taken seriously. Instead of defining the operators $\rho_\alpha$ and $b$ as sums over fermionic bilinear, we could have defined them as $\psi_{0,\alpha}^\dagger \psi_{0,\alpha}$ and $\psi_{0,\uparrow} \psi_{0,\downarrow}$, respectively, instead. Perhaps a bit awkward, but we could then consider susceptibilities of such an operator as a function of $N_f$ and take $N_f \to 0$ to find that at least there, the susceptibility diverges. This corresponds to the susceptibility in the $N_f = 1$ case diverging, when only summing diagrams without internal fermionic loops. We found that after adding all diagrams with fermionic loops with only 2 vertices to this, the divergence of the susceptibilities around (but not at) the QCP are unchanged. This hints that the region of diverging susceptibilities might remain at finite $N_f$. However, this is far from certain since two important effects in the finite $N_f$ theory are missing in the $N_f \to 0, N_f k_F = $ const limit: Firstly, internal fermion loops with more than 2 vertices are missing. Secondly, the scaling of boson momenta obtained after adding Landau-damping means that the curvature term in the fermion propagator becomes relevant for non-planar diagrams below an energy scale $\omega_\beta$. This energy scale is supressed in the double limit, but is finite for finite $N_f$ and may change the long distance behavior of correlators. However, it is possible to suppress this effect to very long distances by having $v_F k_F \gg \lambda^2/c^2$.

Finally, in Section 5 we applied our findings to specific systems of fermions near QCPs to see whether fluctuations induce instabilities in the $N_f \to 0$ limit. We considered charge nematic, ferromagnetic, spin nematic and circulating current transitions, systems close to multiple of these transitions and finally we also considered fermions coupled to an emergent gauge field.

One of the main reasons for studying this kind of model is the unexplained SC dome in the vicinity of nematic transitions in the cuprates and iron pnictides. As mentioned in the introduction, nematic fluctuations may both lead to a breakdown of quasiparticles that limit pairing and they act as an attractive force inducing pairing. Here we found that the latter effect wins and superconductivity is enhanced by nematic fluctuations in the small $N_f$ limit. Furthermore we argued that the gap symmetry is non-universal, it depends on terms neglected in the low-energy limit we are considering.

In the case of a continuous ferromagnetic or spin-nematic transition we found a naked metallic QCP in the case with one fermion orbital. This is interesting since it admits the study of strong NFL physics all the way to $T = 0$. The naked QCP is crucially single orbital and it is thus not amenable to DQMC relying on two-orbitals to cancel the fermion sign-problem. This makes an alternative approach like the small $N_f$ limit currently necessary to study this naked QCP. In the case of two orbitals we find a spin-triplet SC groundstate as also develops in DQMC as temperature is lowered [48]. The results near the rotation and time-reversal symmetry breaking circulating current transition was maybe the most intriguing. Here we find a phase diagram with a SDW/CDW state induced in a finite gap interval at $T = 0$ near the original QCP and suppressed spin-singlet SC.

We made some comparisons to earlier studies on the same systems. Most interesting are perhaps comparisons to DQMC results. While our results are largely consistent in what instabilities we find, it is hard to make sharp comparisons at this stage. The results found here are valid at distances $\gg 1/k_F$ while the sizes of systems amenable to DQMC are still quite limited.

Additionally, the instabilities we find are only at $T = 0$ which is inaccessible to DQMC. One could approach $T = 0$ and compare the growth of correlators but to do this we would need to match the couplings of our field theory to the interactions in the lattice models studied using DQMC. Admittedly, the $N_f \to 0$ limit might be too far from the finite $N_f$ case to warrant this type of comparison. However, DQMC provides an easy way to study the $N_f \to 0$ limit at finite temperature. The acceptance-rejection probability within DQMC depends on the ratio of fermion Green's function determinants and by raising it to the power of $N_f$, one can easily account for different values of $N_f$. That way it should be possible to make contact with the type of results we provide here and DQMC simulations. The benefit is two-fold, firstly it provides a non-trivial way to benchmark DQMC simulations with analytical results. While the analytic results are at $N_f = 0$, it is still at a strongly interacting point not previously studied. Secondly, it gives us a way to measure the effect of tuning $N_f$ that can shed light on whether the phenomena found here are specific to $N_f = 0$ or if they in some form extend to finite $N_f$.

The fact that the $k \ll k_F$ part of the $\langle \rho \rho \rangle$ correlator receives no corrections from interactions is an interesting result. This was also found in [52] and [36] due to cancellations upon symmetrizing fermion loops with density vertices. We have generalized that result by allowing for momentum dependent vertices and finite temperatures. This has prompted further investigations that will be presented in [53].

When adding Landau-damping corrections to the boson with momentum dependent coupling, we found that corrections from non-centrosymmmtric systems seem to disobey the earlier belief [14,15] that only two patches are important. In this case we in fact find that fermions receive relevant corrections from bosons that have relevant corrections from fermions in all patches. This also warrants further investigation.

A surprising find is the curing of the finite temperature IR divergence in perturbation theory since this was not found to be cured in the recent matrix large $N$ treatment of the same model in [41]. It would be interesting to understand whether the additional crossed diagrams cured the divergence, or if there is some other difference between our works.

The framework developed in [27] and used here has proved to be quite convenient for studying quantum critical metals, albeit at $N_f \to 0$. Many results can be found analytically and the calculations are not more difficult than those in one-loop perturbation theory. In particular we find the calculations easier than the much employed matrix large $N$ limit that contains a subset of the diagrams in the small $N_f$ limit. We believe it would be useful to consider this framework further to develop an understanding similar to that of perturbation theory. Specifically it would be interesting to systematically understand how properties of the boson Green's function affects the $h^\pm$ functions and what combinations of $h^\pm$ functions show up in the exponents. For all the coupling functions we considered, and for all the fermion bilinear correlators we considered, we did not find any quasi-long range order and we found that the finite temperature IR divergence was cured. While not manifest, it seems likely that these findings are more generally true, for all couplings and correlators. It would be interesting to investigate whether this is the case, and what happens for other boson Green's functions.

## Acknowledgements

The author wishes to thank Abdelmalek Abdesselam, Chris Hooley, Andriy Nevidomskyy and Johan Schött for useful discussions. This work was supported by the grant "Exact Results in Gauge and String Theories" from the Knut and Alice Wallenberg foundation.

## A $h^\pm$ at the QCP

Calculating $h^\pm$ for $\tau = 0$ at the QCP $T = r = 0$ we find

$$h^+(0,x) = |x| \frac{\sqrt{c^2 - v_F^2} - v_F \cos^{-1}\left(\frac{v_F}{c}\right)}{4\pi c (c^2 - v_F^2)^{3/2}} \tag{132}$$

$$h^-(0,x) = -|x| \frac{\cos^{-1}\left(\frac{v_F}{c}\right)}{4\pi c v_F \sqrt{c^2 - v_F^2}}, \tag{133}$$

for $v_F \neq c$ and for $v_F = c$ we have

$$h^-(0,x) = -\frac{|x|}{4\pi v_F^3} \tag{134}$$

$$h^+(0,x) = \frac{|x|}{12\pi v_F^3}. \tag{135}$$

Despite not manifestly so, these functions are continuous in $v_F$ at $v_F = c$.

## B Calculating $h^\pm$ with a gapped boson at zero temperature

Consider a gapped boson propagator $D$ that after integrating over $k_y$ is continuous at $\omega = 0, k_x = 0$:

$$\int \frac{dk_y}{2\pi} D(0,0,k_y) = A < \infty, \tag{136}$$

and further decays algebraically at large $\omega$ and $k_x$.

We want to calculate the large $\tau, x$ behavior of the corresponding $h^\pm$-functions at $T = 0$. The $h^\pm$ integrals can be viewed as regularized, symmetrized Fourier transforms of the $h^\pm$ integrands without the numerator,

$$h^\pm(\tau,x) = \lim_{\epsilon \to 0} \left( \frac{\mathcal{F}_\epsilon[d^\pm](\tau,-x)}{2} + \frac{\mathcal{F}_\epsilon[d^\pm](-\tau,x)}{2} - \mathcal{F}_\epsilon[d^\pm](0,0) \right), \tag{137}$$

where $\epsilon$ is a cutoff that regularizes the divergence at $\omega = k_x = 0$ and

$$d^\pm(\omega,k_x) = \frac{1}{(i\omega_n - v_F k_x)(-i\omega_n \pm v_F k_x)} \int \frac{dk_y}{2\pi} D(\omega,k_x,k_y). \tag{138}$$

The unregularized Fourier transform diverges for all $\tau, x$ due to the singularity at $\omega = k_x = 0$ but the divergence is uniform so the subtraction of the $\tau = x = 0$ component allows us to remove the regulator $\epsilon$. For any $\epsilon > 0$, we have that $d^\pm$ is absolute square integrable when using the same cutoff and so is its Fourier transform by the Plancherel theorem. This means that the Fourier transform necessarily decays at large $\tau, x$ for any finite $\epsilon$. This means that if the $h^\pm$ functions were to not approach 0 at large $\tau, x$, then the behavior at infinity can be obtained from $d^\pm(\omega,k_x)$ for $\omega^2 + k_x^2 < \epsilon^2$ for any $\epsilon > 0$. We can thus use

$$h^\pm(\tau,x) = \int\limits_{\omega^2 + v_F^2 k_x^2 < \epsilon^2} \frac{d\omega dk_x}{(2\pi)^2} \frac{\cos(\omega_n \tau - k_x x) - 1}{(i\omega_n - v_F k_x)(-i\omega_n \pm v_F k_x)} A + \mathcal{O}\left(\frac{1}{\sqrt{\tau^2 + x^2/v_F^2}}\right), \tag{139}$$

to calculate the large $\tau$, $x$ limit for a gapped $D$ at $T = 0$. We then find:

$$h^+(\tau, x) = \text{finite} \tag{140}$$

$$h^-(\tau, x) = -\frac{\log(v_F^2 \tau^2 + x^2)}{4\pi v_F} A + \text{finite} . \tag{141}$$

Here the finite part depends on how $D(\omega, k_x, k_y)$ behaves away from $\omega = k_x = 0$.

## C   Calculating $h^\pm$ at finite temperature

We calculate the $n = 0$ contribution to the $h^\pm(\tau, x)$ functions at finite temperature for $\phi$ propagator

$$D(\omega_n, k_x, k_y) = \frac{1}{\omega_n^2 + c^2 k_x^2 + c^2 k_y^2 + r} . \tag{142}$$

We find

$$
\begin{aligned}
h^\pm_{n=0}(x) &= \mp T \int \frac{dk_x}{2\pi} \frac{\cos(k_x x) - 1}{v_F^2 k_x^2} \frac{1}{2c\sqrt{r + c^2 k_x^2}} \\
&= \mp T \frac{2 + G^{2,1}_{1,3}\left( \frac{rx^2}{4c^2} \Big| \begin{matrix} \frac{3}{2} \\ 0, 1, \frac{1}{2} \end{matrix} \right)}{4\pi r v_F^2} ,
\end{aligned}
\tag{143}
$$

where $G$ is the Meijer $G$-function. In the large $|x|$ limit we have

$$h^\pm_{n=0}(\tau, x) = \pm \frac{T}{4c v_F^2 \sqrt{r}} |x| + \text{finite} , \tag{144}$$

and taking $r \to 0$ we find

$$h^\pm_{n=0}(\tau, x) = \mp \frac{T x^2}{8\pi c^2 v_F^2} \log\left( \frac{rx^2}{c^2} \right) + \text{finite} . \tag{145}$$

Now we consider the double limit $c/\sqrt{r} \ll x \ll v_F/T$. Using that the $\phi$ Green's function decays monotonically in $|\omega_n|$ we can bound the $h^-$ Matsubara sum with $T = 0$ integrals on both sides shifted by either an extra or a removed $h^-_{n=0}$ to obtain

$$|h^-(\tau, x) - h^-_{T=0}(\tau, x)| < |h^-_{n=0}(x)| . \tag{146}$$

Further we use that

$$0 \leq \int \frac{dk_y}{2\pi} D(\omega_n, k_x, k_y) \leq \frac{1}{2c\sqrt{r}} , \tag{147}$$

to find

$$|h^\pm_{n=0}(x)| < \frac{T|x|}{4c v_F^2 \sqrt{r}} . \tag{148}$$

We thus find that $|h^-_{T=0}(\tau, x)|$ in Eq. (141) is dominating this for $c/\sqrt{r} \ll x \ll v_F/T$ and $h^-(\tau, x) \to h^-_{T=0}(\tau, x)$ in these limits.

# D    Symmetrized loop cancellation

Here we show that to leading order in large $k_F$, fermion loops with 3 or more vertices cancel upon symmetrizing the vertices. We do this at a finite temperature $T$ and allow for momentum dependent vertices. We start off with two lemmas:

**Lemma D.1.** *For $n \in \mathbb{N}$, and $n \geq 2$ we have*

$$\sum_{j=1}^{n} \prod_{i=1, i \neq j}^{n} \frac{1}{z_i - z_j} = 0, \tag{149}$$

*where $z_i \in \mathbb{C}$ and $z_i \neq z_j$ if $i \neq j$.*

*Proof.* Consider a contour $C$ at $\infty$ that surrounds all poles:

$$\int_C \mathrm{d}z \prod_{i=1}^{n} \frac{1}{z - z_i} = 2\pi i \sum_{j=1}^{n} \prod_{i=1, i \neq j}^{n} \frac{1}{z_j - z_i}. \tag{150}$$

The contour integral is 0 for $2 \leq n$. □

**Definition D.1.** *We call a set of complex numbers $Z$ nonexceptional if for every non-empty proper subset $Z_0 \subset Z$ we have*

$$\sum_{z \in Z_0}^{n} z \neq 0. \tag{151}$$

**Lemma D.2.**

$$\sum_{\sigma \in S_n} \sum_{j=1}^{n} \left( \sum_{l=1}^{j} w_{\sigma(l)} \right) \prod_{i=1, i \neq j}^{n} \frac{1}{\sum_{l=1}^{j} z_{\sigma(l)} - \sum_{l=1}^{i} z_{\sigma(l)}} = 0, \tag{152}$$

*where $w_i, z_i \in \mathbb{C}$, $\{z_i\}$ is nonexceptional and $S_n$ is the symmetric group over the first $n$ positive integers.*

See [54] for a proof of this identity. A simplified proof together with a generalization of the following loop cancellation is to be published [53].

Now consider the finite termperature symmetrized fermion loop with $n$ vertices with external momenta $q_i = (\epsilon_i, \mathbf{q}_i)$:

$$I = T \sum_{\sigma \in S_n} \sum_{\omega_m} \int \frac{\mathrm{d}^2 \mathbf{k}}{(2\pi)^2} \prod_{i=1}^{n} G(k + Q_i^{\sigma}) f\left(k + \frac{Q_{i-1}^{\sigma} + Q_i^{\sigma}}{2}\right), \tag{153}$$

where $\omega_m = (2m+1)\pi T$ are fermionic Matsubara frequencies, $k = (\omega_m, \mathbf{k})$, and

$$Q_i^{\sigma} = \sum_{j=1}^{i} q_{\sigma(j)}. \tag{154}$$

By energy and momentum conservation we have $Q_n^{\sigma} = 0$. The function $f$ is the momentum dependent coupling function and it varies between momenta of order $k_F$. The momentum integral is dominated by the region close to the Fermi surface. We restrict the momentum integral to a patch $P$ of the Fermi surface small compared to $k_F$. The coupling function can then be treated as a constant within this patch, with the corrections being next-to-leading order in large $k_F$. We then need to prove that the cancellation of the large $k_F$ leading order

term happens after symmetrizing, but before summing up all different patches. We use patch coordinates: $k_x$ across the Fermi surface and $k_y$ parallel to it and we linearize the Green's function:

$$G_P(k) = \frac{1}{i\omega_m - v_F k_x - v_F k_y^2/2k_F}. \tag{155}$$

First we integrate out $k_x$ with a contour around the upper half-plane.

$$I_P = T \sum_{\sigma \in S_n} \sum_{\omega_m} \int_P \frac{dk_y}{2\pi} \sum_{j=1}^{n} \frac{-i\theta(\omega_m + \Omega_j^\sigma)}{v_F}$$
$$\times \prod_{i=1,i\neq j}^{n} \frac{f_P}{i(\Omega_i^\sigma - \Omega_j^\sigma) - v_F(K_{x,i}^\sigma - K_{x,j}) - v_F((k_y + K_{y,i}^\sigma)^2 - (k_y + K_{y,j})^2)/2k_F}$$
$$+ \text{subleading}, \tag{156}$$

where $\theta$ is the Heaviside step functions. Now the $\omega_m$ sum is trivial. The summand is clearly 0 for $\omega_m < \min_j(-\Omega_j)$. For $\omega_m > \max_j(-\Omega_j^\sigma)$ we find that the $\omega_m$-summand is of the form

$$\sum_{j=1}^{n} \prod_{i=1,i\neq j}^{n} \frac{1}{z_i - z_j}, \tag{157}$$

and we can apply Lemma D.1 to find that it is zero there as well. We can thus restrict the $\omega_m$ sum to some upper cutoff $\omega_N > \max_j(-\Omega_j^\sigma)$. The benefit is that each of the terms in the $j$ sum now converge and we can exchange the orders, perform the now restricted $\omega_m$ sum to obtain

$$I_P = f_P^n \sum_{\sigma \in S_n} \int dk_y \sum_{j=1}^{n} \frac{i\Omega_j^\sigma}{v_F}$$
$$\times \prod_{i=1,i\neq j}^{n} \frac{1}{i(\Omega_i^\sigma - \Omega_j^\sigma) - v_F(K_{x,i}^\sigma - K_{x,j}^\sigma) - v_F((k_y + K_{y,i}^\sigma)^2 - (k_y + K_{y,j}^\sigma)^2)/2k_F}$$
$$+ \text{subleading}. \tag{158}$$

Here we have used Lemma D.1 again to get rid of the upper limit of the primitive function. We note that this is now independent of temperature. Finally, we are interested in the large $k_F$ limit. We cannot exchange the limit with the integral right away since the $k_y$ that are relevant will grow with $k_F$ in the limit. We therefore make the change of variables $k_y \to sk_F$ and we have

$$I_P = \frac{ik_F f_P^n}{v_F} \sum_{\sigma \in S_n} \int ds \sum_{j=1}^{n} \Omega_j$$
$$\times \prod_{i=1,i\neq j}^{n} \frac{1}{i(\Omega_i - \Omega_j) - v_F(K_{x,i} - K_{x,j}) - v_F(K_{y,i}^2 - K_{y,j}^2)/2k_F - v_F s(K_{y,i} - K_{y,j})}$$
$$+ \text{subleading}. \tag{159}$$

Now it is safe to take the large $k_F$ limit of the integrand and we obtain an expression that it is of the form of the sum in Lemma D.2. It thus cancels out to leading order in large $k_F$ upon symmetrizing. Note that this happens before performing the $k_y$ integral. This argument is made more rigorously and is further generalized in [53].

# E  Calculating 1-loop boson polarization

For brevity, we consider spin- and flavorless fermions here. The one-loop boson polarization is given by:

$$\Pi_2[f](\omega_n, \mathbf{q}) = -T \sum_{\omega_m} \int \frac{\mathrm{d}^2 k}{(2\pi)^2} \frac{f(\mathbf{k} + \mathbf{q}/2)}{(i\omega_m - \epsilon(\mathbf{k}) + \mu)(i(\omega_m + \omega_n) - \epsilon(\mathbf{k} + \mathbf{q}) + \mu)} . \tag{160}$$

Performing the Matsubara sum we have

$$\Pi_2[f](\omega_n, \mathbf{q}) = -\int \frac{\mathrm{d}^2 k}{(2\pi)^2} f(\mathbf{k} + \mathbf{q}/2) \frac{\tanh\left(\frac{\epsilon(\mathbf{k}) - \mu}{2T}\right) - \tanh\left(\frac{\epsilon(\mathbf{k} + \mathbf{q}) - \mu}{2T}\right)}{2(i\omega_n + \epsilon(\mathbf{k}) - \epsilon(\mathbf{k} + \mathbf{q}))} . \tag{161}$$

The denominator vanishes at $2\mathbf{k} = -\mathbf{q}$ if $\omega_n = 0$ but the numerator also vanishes there so the above integrand is finite and bounded as long as the coupling function is. We go to polar coordinates $k, \theta$ where $\theta$ is the angle between $\mathbf{k}$ and $\mathbf{q}$. The difference of the hyperbolic tangent functions decays exponentially away from the Fermi surface with decay length $T/v_F$. We can thus linearize the dispersion, coupling function and integral measure around $k = k_F$ for $q, T/v_F \ll k_F$ and additionally we can extend the $k$ integral to all of $\mathbb{R}$:

$$\Pi_2[f](\omega_n, \mathbf{q}) = -\int \frac{\mathrm{d}k\mathrm{d}\theta}{(2\pi)^2} f(\theta) \frac{\tanh\left(\frac{v_F(k - k_F)}{2T}\right) - \tanh\left(\frac{v_F(k - k_F + q\cos(\theta))}{2T}\right)}{2(i\omega_n - v_F q\cos(\theta))} + \text{subleading}. \tag{162}$$

Performing the $k$ integral we find

$$\Pi_2[f](\omega_n, \mathbf{q}) = -\frac{k_F q}{(2\pi)^2} \int \mathrm{d}\theta \frac{f(\theta)\cos(\theta)}{i\omega_n - v_F q\cos(\theta)} + \text{subleading}. \tag{163}$$

We find that this is q independent and generally non-zero for $\omega_n = 0$. This means that interactions generate a boson gap but we have already considered a gap $r$ so we can simply absorb this in a redefinition of $r$. We continue to work with the polarization with this constant subtracted and define:

$$\tilde{\Pi}_2[f](\omega_n, \mathbf{q}) = -\frac{i k_F \omega_n}{(2\pi)^2 v_F} \int \mathrm{d}\theta \frac{f(\theta)}{i\omega_n - v_F q\cos(\theta)} . \tag{164}$$

We now expand the function $f$ in Fourier series according to Eq. (112) and discard the $\sin(\theta)$ terms since they are odd under the $\theta$ integral:

$$\tilde{\Pi}_2[f](\omega_n, \mathbf{q}) = -\frac{i k_F \omega_n}{(2\pi)^2 v_F} \int \mathrm{d}\theta \frac{1}{i\omega_n - v_F q\cos(\theta)} \sum_{m=0}^{\infty} f_m \cos(m\theta) . \tag{165}$$

We write the fraction as a geometric series to obtain

$$\tilde{\Pi}_2[f](\omega_n, \mathbf{q}) = -\frac{k_F}{(2\pi)^2 v_F} \sum_{l=0,m=0}^{\infty,\infty} \int \mathrm{d}\theta \left(\frac{v_F q\cos(\theta)}{i\omega_n}\right)^l f_m \cos(m\theta) . \tag{166}$$

Here we strictly have to assume $v_F q < |\omega_n|$, but we later see that we can extend the result through analyticity in $\omega_n$. We expand the cosine power using the binomial theorem and finally the $\theta$ integral is trivial.

$$\int \mathrm{d}\theta \cos(\theta)^l \cos(m\theta) = 2^{-1-l} \int \mathrm{d}\theta \sum_{k=0}^{l} \binom{l}{k} e^{i(2k-l)\theta} (e^{im\theta} + e^{-im\theta}) \tag{167}$$

$$= \begin{cases} 2^{-l}\pi\left(\binom{l}{\frac{l+m}{2}} + \binom{l}{\frac{l-m}{2}}\right) & \text{if } l \geq |m| \text{ and } m \equiv l \mod 2 \\ 0 & \text{otherwise} \end{cases} . \tag{168}$$

Finally we perform the $l$ sum to obtain:

$$\tilde{\Pi}_2[f](\omega_n, \mathbf{q}) = -\frac{k_F|\omega_n|}{2\pi v_F\sqrt{\omega_n^2 + q^2 v_F^2}} \sum_{m=0}^{\infty} f_m\left(\frac{-i\,\text{sgn}(\omega_n)v_F q}{\sqrt{\omega_n^2 + q^2 v_F^2} + |\omega_n|}\right)^m. \tag{169}$$

This is manifestly analytic in $\omega_n$ away from the imaginary axis so we may use this also for $v_F q > |\omega_n|$.

## F  Real space resummed 1-loop boson polarization

Here we calculate the Fourier transform of the resummed bubbles of Eq. (110).

$$\langle \rho\rho(x)\rangle_{\text{bubbles}} = \frac{1}{\lambda^2}\int\frac{\mathrm{d}\omega\mathrm{d}^2 q}{(2\pi)^3}e^{-i\omega\tau + i\mathbf{q}\cdot\mathbf{x}}\sum_{n=1}^{\infty}\tilde{\Pi}_2[\lambda^2](\omega, q)^n D(\omega, q)^{n-1}. \tag{170}$$

Here $\Pi_2(\omega, k)$ is the 2-vertex fermion loop with density vertices, see Eq. (113). Instead of performing the geometric sum we do the Fourier transform term by term. Going to polar coordinates for the spatial integral and performing the angular integral we have

$$\langle \rho\rho(x)\rangle_{\text{bubbles}} = \frac{1}{\lambda^2}\int\frac{\mathrm{d}\omega\mathrm{d}q}{(2\pi)^2}qJ_0(qx)e^{-i\omega\tau}\sum_{n=1}^{\infty}\left(-\frac{\lambda^2 k_F|\omega|}{2\pi v_F\sqrt{\omega^2 + q^2 v_F^2}}\right)^n\frac{1}{(\omega^2 + c^2 q^2 + r)^{n-1}}, \tag{171}$$

where $J_0$ is the zeroth Bessel function of the first kind. Considering $|\tau| \gg r^{-1}$ and $x \gg v_F r^{-1}$ we are only interested in energies and momenta $k \ll r v_F^{-1}$ and $|\omega| \ll r$ and we can approximate the boson propagator as $1/r^2$. We can then perform the $k$ integral

$$\langle \rho\rho(x)\rangle_{\text{bubbles,LR}}(\tau, x) = \int\frac{\mathrm{d}\omega}{2\pi}e^{-i\omega\tau}\sum_{n=1}^{\infty}\frac{2^{-\frac{n}{2}}nm^2\left(-\frac{M_D^2}{m^2}\right)^n\left(\frac{x\omega}{v_F}\right)^{\frac{n+2}{2}}K_{\frac{n}{2}-1}\left(\frac{x\omega}{v_F}\right)}{\lambda^2 x^2\Gamma\left(\frac{n}{2}+1\right)}, \tag{172}$$

where

$$M_D^2 = \frac{N_f k_F \lambda^2}{2\pi v_F}. \tag{173}$$

Finally we perform the $\omega$ integral and the sum in $n$

$$\langle \rho\rho(x)\rangle_{\text{bubbles,LR}}(x) = \frac{v_F r}{2\pi^{3/2}\lambda^2 x^2 X^3}\sum_{n=0}^{\infty}\left(-\frac{M_D^2 x}{rX}\right)^n\left(x^2 - n\tau^2 v_F^2\right)\frac{\Gamma\left(\frac{n+1}{2}\right)}{\Gamma\left(\frac{n}{2}\right)} \tag{174}$$

$$= \frac{r^2 M_D^2 v_F\left(2x^2 X^2\left(M_D^4 - r^2\right) + r^2 X^4 - M_D^4 x^4\right)}{2\pi^2\lambda^2 x X^2\left(M_D^4 x^2 - r^2 X^2\right)^2} \tag{175}$$

$$+ \frac{r^2 M_D^4 v_F\left(r^2\left(x^2 - 2\tau^2 v_F^2\right) - M_D^4 x^2\right)\cos^{-1}\left(\frac{M_D^2 x}{rX}\right)}{2\pi^2\lambda^2\left(r^2 X^2 - M_D^4 x^2\right)^{5/2}}, \tag{176}$$

where

$$X = \sqrt{x^2 + v_F^2\tau^2}. \tag{177}$$

This decays as $(x^2 + v_F^2\tau^2)^{-3/2}$ at large $x$ and $\tau$.

# G   Calculating $h^\pm$ with Landau-damping corrections

We want to calculate the $h^\pm$ functions with a corrected boson propagator

$$D_{\text{LD}}(q) = \frac{1}{\omega^2 + c^2(q_x^2 + q_y^2) + r - \Pi_2[\lambda(\theta)^2](q)}, \tag{178}$$

with $\Pi_2[\lambda(\theta)^2](q)$ given by Eq. (113). For $T = 0$, $0 < r$ we can use the result of Appendix B since $D$ is gapped and still decays at least algebraically for large energies and momenta. We proceed to finite temperatures. Eq. (97) still applies and we need only consider the $n = 0$ contribution. However, the boson self-energy is 0 for $\omega_n = 0$ so there is no difference from the non-Landau damped case apart from in the finite part.

Finally we consider the QCP. Here we use a simplified propagator

$$D_{\text{IR}}(q) = \frac{1}{c^2 q_y^2 + M_D^2 \frac{|\omega|}{q_y}}, \tag{179}$$

motivated by the low-energy scaling in Eq. (114) to find the large $x$ limit ($y = 0$ so we neglect $q_x$ above) and verify the result using numerics and the full propagator. We find:

$$h_+^{\text{LD}}(0, x) = -|x|^{1/3} \frac{\Gamma\left(\frac{2}{3}\right)}{3\sqrt{3}\pi M_D^{2/3} v_F^{4/3}} + \text{finite} \tag{180}$$

$$h_-^{\text{LD}}(0, x) = -3 h_+^{\text{LD}}(0, x) + \text{finite}, \tag{181}$$

at the QCP.

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
