# Peer review of "Instabilities of quantum critical metals in the limit $N_f\rightarrow0$"

_SciPost Physics, doi:SciPost Phys. 10, 067 (2021)_

## Round 1 · Referee Report · Ipsita Mandal (Referee 1) · 2020-11-17

Strengths

1) The author has investigated instabilities of fermions in two dimensions in presence of order parameter fluctuations near different QCPs and emergent gauge fields in the Nf→0 limit, where Nf is the number flavours.

2) The computations involved both zero and nonzero temperatures.

Weaknesses

1) The physical significance of Nf->0 limit is not clear to the reader.

2) It is not clear why this method has any advantage, if any, over other methods like dimensional regularization, or where the dispersion of the of the boson is modified (e.g. Ref. 21 in the references).

Report

I would recommend the publication of the paper provided the author addresses the questions and improvements listed below.

Requested changes

1) I would recommend the author explains the Nf->0 limit intuitively.

2) What is the significance of keeping Nf kF = constant? Does this have any relation to the fact the in 2+1 d critical fermi surface systems, one can use patch theory as the physical observables are effectively local?

3) While the author mentions Ref. 21 in the context of Cooper pairing for the Ising-nematic case, the reference --"Superconducting instability in non-Fermi liquids, Ipsita Mandal, Phys. Rev. B 94, 115138 – Published 16 September 2016" is missing.

4) It would be good if the author explains the usefulness of his method compared to those used in Ref. 10, or Ref. 21. Relation to and citations for other relevant prior works are missing, e.g., Phys. Rev. B 91, 125136 (2015) , Eur. Phys. J. B (2016) 89: 278, Annals of Physics, 376, 89 (2017), Phys. Rev. X 7, 021010 (2017), Phys. Rev. B 98, 024510 (2018), arXiv:2006.10766. I would like to see some discussions pointing out the advantage or disadvantage of the author's formalism compared to the ones used in these papers.

5) The author mentions that "IR divergence in perturbation theory at finite temperatures for critical fluctuations was cured by summing all diagrams in the small Nf limit." Is there any concrete mathematical justification why this is the case?

6) The author comments that "We found that after adding all diagrams with fermionic loops with only 2 vertices to this, the divergence of the susceptibilities around (but not at) the QCP are unchanged. If that is any indication as to what happens when adding further finite Nf corrections in terms of diagrams with fermionic loops with more than 2 vertices, we would expect to find the same divergences of susceptibilities at finite Nf. However, this indication may be deceiving and these further corrections may cancel out or dominate the divergence found here." I wonder what is the takeaway message here. What is the proof or even intuition that what the author has done is a controlled approximation?

  • validity: good
  • significance: ok
  • originality: ok
  • clarity: low
  • formatting: good
  • grammar: good

Author:  Petter Säterskog  on 2021-01-28  [id 1186]

(in reply to Report 1 by Ipsita Mandal on 2020-11-17)

Dear Prof. Ipsita Mandal, Thank you for your comments on my manuscript. I have copied your requested changes and make replies to them below:

1) I would recommend the author explains the Nf->0 limit intuitively.

Petter: I have added some remarks explaining the intuition in the introduction.

2) What is the significance of keeping Nf kF = constant? Does this have any relation to the fact the in 2+1 d critical fermi surface systems, one can use patch theory as the physical observables are effectively local?

Petter: I do not believe this is strictly related. The large kF limit indeed means that we can consider the patch theory (or the pair of antipodal patches for 4 point functions). However, what we are after here is the cancellation that happens for fermion loops with more than two vertices upon symmetrization. Fermion loops are all proportional to NfkF but the leading term cancels upon symmetrization of the vertices for loops with more than 2 vertices. They are thus proportional to Nfk_F^p, with p<1 and the boson remains Gaussian when taking this limit since corrections are only to the two-point function. This is described in section 4 but I have expanded this a bit and also added some more comments about it in the introduction and in the discussion. To conclude, it is not that this limit is particularly interesting from a physics viewpoint, the reason we use it is because we can add 2 vertex fermion loops easily to get a more physical result, and this limit is a way to control that approximation.

3) While the author mentions Ref. 21 in the context of Cooper pairing for the Ising-nematic case, the reference --"Superconducting instability in non-Fermi liquids, Ipsita Mandal, Phys. Rev. B 94, 115138 – Published 16 September 2016" is missing.

Petter: Thank you for bringing this to my attention, indeed this warrants a mention as well and has been added.

4) It would be good if the author explains the usefulness of his method compared to those used in Ref. 10, or Ref. 21. Relation to and citations for other relevant prior works are missing, e.g., Phys. Rev. B 91, 125136 (2015) , Eur. Phys. J. B (2016) 89: 278, Annals of Physics, 376, 89 (2017), Phys. Rev. X 7, 021010 (2017), Phys. Rev. B 98, 024510 (2018), arXiv:2006.10766. I would like to see some discussions pointing out the advantage or disadvantage of the author's formalism compared to the ones used in these papers.

Petter: Phys. Rev. X 7, 021010 (2017) studied the antiferromagnetic QCP where the critical fluctuations are at a finite Q. This is thus a different kind of model from what I study (as mentioned in the introduciton I only consider Q=0). In studying their model they find an emergent small parameter that allows them to self-consistently find critical exponents. I am not aware if something similar is possible for the Q=0 quantum critical metals. If it were, then I believe my method would be disadvantaged by having an artificial small parameter Nf as opposed to the emergent small parameter v. The advantage of this work would be that we can perform the calculation at finite temperature and for a gapped phi and find explicit correlation functions.

All of: Ref 10, Ref 21, Phys. Rev. B 91, 125136 (2015), Eur. Phys. J. B (2016) 89: 278, Annals of Physics, 376, 89 (2017), Phys. Rev. B 98, 024510 (2018), and arXiv:2006.10766 have departed from the theory of interest here by changing the dimensionality in different ways such that it is no longer strongly coupled. This allows the authors to study the model perturbatively and find the renormalization group flows. My work has also departed from the theory of interest by taking Nf->0 but it is crucially still strongly coupled and an infinite set of complex diagrams has to be summed to calculate correlation functions. The epsilon expansions can find new fixed points but strong coupling makes it impossible to tell if they persist down to 2+1 dimensions and what their scalings would be there. Similarly my work gives correlation functions at Nf=0, but it is ultimately unclear what resemblance these bear to the Nf=1 case. The advantage of the small Nf approach is that it actually works at strong coupling and additionally that it gives explicit correlation functions at 0 and finite T. I also think the intuition is a bit clearer in the small Nf case than in the case of generalized dimensions of the Fermi surface or its embedding space.

I have added a few of these as additional references for the mention of works in generalized dimensions and expanded a bit on the comparison with my work to these in the introduction.

5) The author mentions that "IR divergence in perturbation theory at finite temperatures for critical fluctuations was cured by summing all diagrams in the small Nf limit." Is there any concrete mathematical justification why this is the case?

Petter: By calculating the first perturbative correction to the fermion two-point function we find an IR divergence as we take the gap r->0. We know that if we calculate the two-point function to any finite order in the coupling constant, then it can impossibly be finite as we take r->0 since it is a polynomial in lambda with the lambda^2 term diverging. Calculating the full sum of diagrams results in an expression that is finite as we take r->0: the divergence was cured by summing all diagrams in the small Nf limit. This is the justification for this statement.

Perhaps something else was requested here? Mathematically this is not very interesting, it is similar to the sum

sum_{n=0}^N 1/(-r)^n/n!

diverging as we take r->0 for any finite N, but if we first take N->\infty we get exp(-1/r) which does not diverge as we take r->0 (from above).

6) The author comments that "We found that after adding all diagrams with fermionic loops with only 2 vertices to this, the divergence of the susceptibilities around (but not at) the QCP are unchanged. If that is any indication as to what happens when adding further finite Nf corrections in terms of diagrams with fermionic loops with more than 2 vertices, we would expect to find the same divergences of susceptibilities at finite Nf. However, this indication may be deceiving and these further corrections may cancel out or dominate the divergence found here." I wonder what is the takeaway message here. What is the proof or even intuition that what the author has done is a controlled approximation?

Petter: The take-away message here is simply that the conclusions drawn in the strict Nf->0 limit are unchanged when adding the Landau-damping corrections. In addition to the take-away message, I made a note that this is a good sign if we hope that the Nf=0 results generalize to finite Nf, but note that we should be careful. This was not very clear so I have rewritten this now, and also added a note about curvature corrections also changing the results at finite Nf.

The proof that we obtain the correct correlation functions when taking the strict limit Nf->0 is in Section 2. The proof is constructive, we solve the bavkground field Green's function and perform the path integrals to find the correlation functions.

The theory contains several limits and their ordering is important. Here the Nf->0 limit is taken before the IR limit kF->inf. We have thus proved that this result is controlled by the small Nf limit and the IR limit, taken in this order.

When adding Landau damping, our results are controlled by a different limit, Nf->0 with NfkF a constant, and this limit is taken before performing the integrals over boson momenta and the sums of all diagrams. We provide a proof that fermion loops with more than two vertices are suppressed, after they are symmetrized, in this limit in the appendix. We may then use the previous framework with a modified boson that contains all 2-vertex fermion loop corrections. While it is neat, the limit we use to control this might not add so much and it may be clearer to just consider the effect: we add all diagrams with fermion loops with at most 2 vertices.

Clarity: Low Was there some section or something in particular that could be improved?

---

## Round 1 · Referee Report · Anonymous (Referee 2) · 2020-11-22

Strengths

1 - analytical predictions on stability of coupled fermion-boson systems 2-unexpected deviations from previously calculated results 3- shows the computational strength of the $N_f \to 0$ limit.

Weaknesses

1 - physical interpretation of results is not clear
2 - paper is not self contained, requires referring to author's previous works.

Report

Summary: the paper studies instabilities of fermion-boson models near a quantum critical point (QCP). The paper expands on previous results by the author and collaborators regarding taking the number of fermion flavors $N_f \to 0$, so as to trace out the boson degrees of freedom exactly. In real space this method allows for analytically tractable expressions of various correlation functions, on which the author performs linear stability analysis. The author finds some intriguing deviations from previous results on the same model with other computation schemes.

Recommendation: The paper includes new material and interesting results and is definitely worthy of publication in some form. However, my understanding is that the paper is a continuation of a series of previous papers (Refs. 23, 24,29). Ref. 24 was already published in SciPost. Hence, it does not fulfill one of the “expectations” of SciPost Physics that are necessary for acceptance. I think the paper is better suited for Phys. Rev. B. or similar journals.

Requested changes

1) My first comment regards the structure of the paper. While I trust that the author has correctly performed the calculations, the real-space representation of results is very difficult (for me) to follow. For example Eqs. 37-39 are at the core of the paper’s analysis but are almost impossible to decipher. In addition, statements like “following the reasoning in [24]…” after Eq. 31 make things even harder to follow. The paper is long, so I believe it should be fully self-contained, and I think some of the derivations that appear in previous works by the author and collaborators can be included, at least in the appendices.

2) There are several results which depart from previous analysis of the model, e.g., a) No IR divergence at finite T, differently from Refs [36, 32], and several not cited works e.g. Luca Dell’Anna and Walter Metzner, Phys. Rev. B 73, 045127. b) A naked QCP for the ferromagnetic model. However there is no identification of what is the mechanism that leads to the different results, which is a problem since the analytic expressions themselves aren’t very transparent as I already discussed. Could the author please elaborate on the results and why they should be trusted?

3) To address 1) + 2) I recommend adding a summary of the major results to the introduction, with a physical explanation of e.g. the functions h^\pm and how they determine the results. It may be helpful to compare with a momentum space treatment though that might be a lot of work.

4) In the ferromagnetic case, a large number of previous works have found both first order and finite q transitions. Could the author please comment on this? See e.g. Rev. Mod. Phys. 88, 025006, Phys. Rev. B 79, 075112 and refs. within.

5) It is known that the two limits of the one loop polarization bubble don’t commute, and depend on the order of integration in the bubble (first frequency or first momentum), such that $\Pi(\omega_m = 0, q) = const, \Pi(\omega_m, q=0) = 0$. When working in momentum space this is not so important as it’s generally clear what is the correct limit to be working at. However, it seems that in the real space representations the integration is over the entire frequency/momentum. This e.g. means that the mass r is actually dependent on distance. Could the author clarify how this is accounted for in the calculations?

6) Some minor points:
a. I couldn’t find a definition of $\lambda^\pm$ (Eq. 37 and on) despite that they are critical for the rest of the discussion. Maybe I missed it in which case I apologize.
b. Footnote 1: as noted above, many previous treatments do find that the $\phi^4$ term is relevant except at $T=0$.
c. I disagree with the author’s interpretation of the results of Refs 40,41 (page 27 paragraph 2). The reason why the two works have different results regarding pairing vs NFL is because they are working at very different couplings. See Xiao Yan Xu, …, npj Quantum Materials volume 5, Article number: 65 (2020) where the two papers are reconciled.

  • validity: good
  • significance: good
  • originality: good
  • clarity: ok
  • formatting: reasonable
  • grammar: good

Author:  Petter Säterskog  on 2021-01-28  [id 1187]

(in reply to Report 2 on 2020-11-22)

Dear referee, Thank you for your comments. I have copied them below and added replies after each of them:

Recommendation: The paper includes new material and interesting results and is definitely worthy of publication in some form. However, my understanding is that the paper is a continuation of a series of previous papers (Refs. 23, 24,29). Ref. 24 was already published in SciPost. Hence, it does not fulfill one of the “expectations” of SciPost Physics that are necessary for acceptance. I think the paper is better suited for Phys. Rev. B. or similar journals.

Petter: I do not agree with it only being a part of a series of papers, and because of this not meeting the expectation of a SciPost Physics publication. As all research, this paper builds upon previous work and in this case some of the previous work has been done by me as well. I believe the paper should be judged by its contents, without regards as to who wrote it. As such, I think it is fulfilling the expectations of SciPost Physics, in particular I think it fulfils expectation 1:

"Detail a groundbreaking theoretical/experimental/computational discovery;"

It is true that the N_f->0 limit has been used by me (and several other authors) before. So has the matrix large N limit, vector large N_f, dimensional regularization, DQMC, etc, in many works on the challenging strongly coupled quantum critical metals. In comparison to e.g. the matrix large N limit, even though the N_f->0 limit is a more powerful method (contains a superset of the diagrams, gives explicit correlators with relative ease), it has not seen as many publications as the matrix large N limit on this topic.

Ref 24 (numbering from the first version) develops a framework for calculating general correlation functions in a quantum critical metal in the small N_f limit. This is restricted to: * Zero temperature * Gapless boson * Correlators with large Euclidean time separations * v_F=c * Momentum independent coupling * Spinless fermions * No Landau-damping corrections Ref 24 subsequently only calculates the density-density correlator and only for the Ising-nematic model. It presents this framework, but does not have much in terms of results apart from the exponential decay of Friedel oscillations at T=0.

In the paper under review we extend this framework to: * Zero and finite temperatures * Gapless and gapped boson * General v_F/c * To work with momentum dependent coupling functions * We show that the results are valid also for \tau=0 * Spinful fermions and multicomponent order parameters/multiple QCPs * We add Landau-damping corrections and discuss some subtleties showing up in the double limit N_f\rightarrow0 with N_fk_F constant. We further use this significantly more useful framework to calculate both charge/spin correlation functions and the pair correlation function and we do this for several physical systems where this model can now be applied: charge and spin nematic QCPs, ferromagnetic QCPs, circulating current QCPs, and U(1) gauge field. We find that the previously found exponential decay can be made into an exponential growth for different coupling functions and correlators from that in ref 24. I think this is a groundbreaking discovery. While in hindsight not so surprising given the result in ref 24, it was not realized there and has not been published before.

Given this finding, we then explore what it means: instabilities, and investigate how susceptibilities depend on temperature and the gap and find new power-laws. Another "groundbreaking discovery" is the induced QCP to a charge/spin density wave state at a finite critical order parameter fluctuation gap.

We consider the gapless boson at finite temperature, in which case perturbation theory has an IR divergence. We show that this is cured by the infinite sum of diagrams contained in the small N_f limit by finding a now finite result as the gap is removed. The corresponding finding but in the matrix large N limit, is the topic of ref 30 and 36.

We further investigate Landau damping corrections and make two surprising findings: * These corrections do not affect the asymptotics, except right at the QCP * The earlier wisdom that only antipodal patches have relevant interactions (ref 13) is violated for non-centrosymmetric couplings. This wisdom is used in many works so this finding should be further investigated, i.e. it is breaking new ground.

Finally, we find new inequalities showing how fluctuations from different QCP interact and can suppress each other's instability inducing effects.

Perhaps ref 24 did not have a groundbreaking result, though it had useful calculations, but I believe I have showed here how this work has several groundbreaking results.

Requested changes 1) My first comment regards the structure of the paper. While I trust that the author has correctly performed the calculations, the real-space representation of results is very difficult (for me) to follow. For example Eqs. 37-39 are at the core of the paper’s analysis but are almost impossible to decipher. In addition, statements like “following the reasoning in [24]…” after Eq. 31 make things even harder to follow. The paper is long, so I believe it should be fully self-contained, and I think some of the derivations that appear in previous works by the author and collaborators can be included, at least in the appendices.

Petter: Thank you for your input, I have added the complete derivation of Eqs. 37-39 using the method of [24] and also copied the $k$ and $\eta$ integration from [24] and added it before Eq 31. I agree that this is better and the meaning of the h^\pm functions is now easier to explain.

2) There are several results which depart from previous analysis of the model, e.g., a) No IR divergence at finite T, differently from Refs [36, 32], and several not cited works e.g. Luca Dell’Anna and Walter Metzner, Phys. Rev. B 73, 045127. b) A naked QCP for the ferromagnetic model. However there is no identification of what is the mechanism that leads to the different results, which is a problem since the analytic expressions themselves aren’t very transparent as I already discussed. Could the author please elaborate on the results and why they should be trusted?

Petter: 2a) Thanks for pointing out this work, I have added a reference to it. The work by Dell’Anna and Metzner and [32] are a bit different from [36, 30] and my work in that they consider a model with a Pomeranchuk instability explicitly and derive a gap for the bosonic field that varies with temperature, but is nonzero at finite temperature. [36, 30] and my work instead consider a model that has a bosonic field with a gap set to 0 by hand, also at finite temperature. This results in an IR divergence at the perturbative level. The case of $\xi_0/\xi=0$ at finite T is not considered in the work by Dell’Anna and Metzner as far as I can tell. However, one sees that their 1-loop fermion self-energy (their eq. 42) and the self-consistent self-energy (their eq. 71) would also diverge if their gap was explicitly set to 0 at finite T. Similarly in [32].

I have expanded on the comparison of my results to those of [36] in section 3.4 and now think that they might actually be compatible. 2b) I hope that I have made the analytical expressions more clear with the added derivations in the updated version of the manuscript.

3) To address 1) + 2) I recommend adding a summary of the major results to the introduction, with a physical explanation of e.g. the functions h^\pm and how they determine the results. It may be helpful to compare with a momentum space treatment though that might be a lot of work.

Petter: I have added a summary to the introduction with a physical explanation h^\pm. I skipped a momentum space treatment though that would be interesting to look into.

4) In the ferromagnetic case, a large number of previous works have found both first order and finite q transitions. Could the author please comment on this? See e.g. Rev. Mod. Phys. 88, 025006, Phys. Rev. B 79, 075112 and refs. within.

Petter: I am aware of the first order ferromagnetic transitions but as mentioned in the first of these references, the transition may be second order in disordered systems. A second order transition is found QMC studies: PhysRevX.7.031058, Quantum Materials volume 5, Article number: 65 (2020) I think studying the effect of fluctuations near a second order ferromagnetic QCP is interesting, unless such a transition is definitively ruled out. Whether certain experimental systems are first or second order is beyond the scope of this paper.

5) It is known that the two limits of the one loop polarization bubble don’t commute, and depend on the order of integration in the bubble (first frequency or first momentum), such that Π(ωm=0,q)=const,Π(ωm,q=0)=0. When working in momentum space this is not so important as it’s generally clear what is the correct limit to be working at. However, it seems that in the real space representations the integration is over the entire frequency/momentum. This e.g. means that the mass r is actually dependent on distance. Could the author clarify how this is accounted for in the calculations?

Petter: 5) I am aware of the subtlety in the polarization bubble, it is conditionally convergent and this results in an ambiguity by a constant. However, we have tuned our model to a particular r so this constant is just absorbed in a redefinition of r and we do not have to consider a particular order. We actually perform the integral in the "opposite" order in the appendix because it was easier and have to do this redefinition there. Later we use the boson propagator with this polarization to go to real space when we perform the integrals in the h^\pm functions. However, these integrals are absolutely convergent so we do not find any ambiguity here. How do you mean that the mass r depends on distance because I calculate correlators in real space?

6) Some minor points: a. I couldn’t find a definition of λ± (Eq. 37 and on) despite that they are critical for the rest of the discussion. Maybe I missed it in which case I apologize.

Petter: Indeed it seems to be missing and has now been added. Thanks for pointing it out.

b. Footnote 1: as noted above, many previous treatments do find that the ϕ4 term is relevant except at T=0.

Petter: Thanks, I rephrased this footnote and the text referring to it.

c. I disagree with the author’s interpretation of the results of Refs 40,41 (page 27 paragraph 2). The reason why the two works have different results regarding pairing vs NFL is because they are working at very different couplings. See Xiao Yan Xu, …, npj Quantum Materials volume 5, Article number: 65 (2020) where the two papers are reconciled.

Petter: Thanks for bringing this work to my attention. I've removed this interpretation and the perceived disagreement with my results is then resolved and I have removed the comment about it at the end of the section.

---

## Round 2 · Referee Report · Ipsita Mandal (Referee 1) · 2021-2-12

Report

The author has made sufficient efforts to address all referee comments. Although I do not quite agree with all his arguments, I believe the work merits publication.

In particular, the author comments:"All of: Ref 10, Ref 21, Phys. Rev. B 91, 125136 (2015), Eur. Phys. J. B (2016) 89: 278, Annals of Physics, 376, 89 (2017), Phys. Rev. B 98, 024510 (2018), and arXiv:2006.10766 have departed from the theory of interest here by changing the dimensionality in different ways such that it is no longer strongly coupled. This allows the authors to study the model perturbatively and find the renormalization group flows. My work has also departed from the theory of interest by taking Nf->0 but it is crucially still strongly coupled and an infinite set of complex diagrams has to be summed to calculate correlation functions. The epsilon expansions can find new fixed points but strong coupling makes it impossible to tell if they persist down to 2+1 dimensions and what their scalings would be there. Similarly my work gives correlation functions at Nf=0, but it is ultimately unclear what resemblance these bear to the Nf=1 case. The advantage of the small Nf approach is that it actually works at strong coupling and additionally that it gives explicit correlation functions at 0 and finite T. I also think the intuition is a bit clearer in the small Nf case than in the case of generalized dimensions of the Fermi surface or its embedding space."

I totally disagree with these arguments. Dimensional regularization is a well-defined technique backed by the concrete proofs that as the spatial dimensions of a system are increases, quantum fluctuations weaker, and as such thee is a definition of upper critical dimension. This was the main idea of Wilson's \phi^4 theory where this was implemented. But the author's method involves taking N_f->0, which is a singular limit, and there cannot me a concrete mathematical justification for this singular limit. This reminds me of the replica trick where people take number of replicas n ->0 at the end, which gives correct results in most cases, but this step is not mathematically justified.

With this grain of salt, I accept this as a method which happens to work for some systems.

---

## Round 2 · Referee Report · Anonymous (Referee 2) · 2021-2-25

Report

The author has addressed my concerns satisfactorily. Regarding the innovation of the work, the author is correct that while instabilities have been studied extensively in other formalisms, it was not done in the real-space formulation. So, I retract my previous objection to publication in SciPost and am happy to recommend publication.

---

## Round 2 · List of Changes

• Added some intuitive description of the Nf->0 limits under consideration to the introduction.
  • Added a summary of the main results to the Introduction
  • Added the complete derivation of Eq. 60, 61 instead of referring to previous work.
  • Added a comment about quantum/thermal contributions and some more references to earlier works
  • Added some references to earlier works that were missed.
  • Added some discussion of the IR divergence in ref 41.
  • Expanded on introduction to Section 4
  • Added some comments on Fermi surface curvature corrections in Section 4 and in the Discussion.
  • Added a comment about temperature dependent boson gap and reference to earlier works.
  • Added some physical explanation of the h^\pm functions to the summary of results in the introduction, to Section 2, and to the Discussion.
  • Reformulated the discussion of what can be expected at finite Nf in the Discussion.

---

## Editorial Decision

published